ecology, genomics, health and disease and epidemiology

sea star wasting, RNA-sequencing, collagen, microbiome, innate immunity, transcriptome

**Author for correspondence:**
Melissa H. Pespeni
e-mail: mpespeni@uvm.edu

# Sea stars resist wasting through active immune and collagen systems

Melissa H. Pespeni and Melanie M. Lloyd

Department of Biology, University of Vermont, 109 Carrigan Drive, Burlington, VT 05405, USA

MHP, 0000-0001-5447-6678

Epidemics are becoming more common and severe, however, pinpointing the causes can be challenging, particularly in marine environments. The cause of sea star wasting (SSW) disease, the ongoing, largest known panzootic of marine wildlife, is unresolved. Here, we measured gene expression longitudinally of 24 adult *Pisaster ochraceus* sea stars, collected from a recovered site, as they remained asymptomatic (8 individuals) or naturally progressed through SSW (16 individuals) in individual aquaria. Immune, tissue integrity and pro-collagen genes were more highly expressed in asymptomatic relative to wasting individuals, while hypoxia-inducible factor 1-α and RNA processing genes were more highly expressed in wasting relative to asymptomatic individuals. Integrating microbiome data from the same tissue samples, we identified genes and microbes whose abundance/growth was associated with disease status. Importantly, sea stars that remained visibly healthy showed that laboratory conditions had little effect on microbiome composition. Lastly, considering genotypes at 98 145 single-nucleotide polymorphism, we found no variants associated with final health status. These findings suggest that animals exposed to the cause(s) of SSW remain asymptomatic with an active immune response and sustained control of their collagen system while animals that succumb to wasting show evidence of responding to hypoxia and dysregulation of RNA processing systems.

## 1. Introduction

Mass mortality events are increasing in frequency and severity for wildlife populations [1–3]. However, the causes, particularly in the marine environment, are challenging to pinpoint [4–6]. Such mass mortality events could be driven by emerging infectious diseases, environmental perturbations or both. For example, changes in abiotic or biotic factors such as temperature, climate variability, ocean acidification, pollutants, pathogens or invasive species can have cascading effects that reduce ecosystem functioning and make populations or individuals more vulnerable to disease [4,7–9]. Given this complexity, integrative approaches are needed to understand the potential factors driving disease emergence [5,6,9,10].

Sea star wasting (SSW) disease is considered the largest panzootic of marine wildlife in history due to its geographical spread from Alaska to Mexico, the greater than 20 species affected and high mortality (100% for some species at some sites) [11–13]. Diseased sea stars present with lesions that progressively worsen; their bodies lose turgor pressure and detach from surfaces; and their arms twist or walk away from the rest of the body. Death and disintegration occur within 2–14 days, leaving a silhouette of white organic matter and ossicles. The mass mortality of these keystone predators has had major ecological consequences [12–15]. For example, the loss of predation on mussels and urchins has led to the alteration of rocky intertidal habitat and the decimation of kelp forests, respectively, threatening coastal biodiversity and ecosystems [14,16,17].

The cause of the disease, however, is unresolved [18–20]. Factors often associated with outbreaks, such as host density and elevated temperatures, have had mixed associations with SSW depending on the site, species and data sources [12,13,21–23], suggesting that wasting may be caused by a combination of biological and environmental factors [13,20,24]. Recent work has suggested that copiotrophic microbial assemblages (microbes that thrive in environments rich with organic matter and low in oxygen [25]) proliferate before SSW onset and deplete oxygen availability at the animal–water interface, impeding sea star respiration and leading to damage and decomposition of tissue [19].

Integrative genomic approaches can be used to implicate the potential causes of disease emergence and health in the oceans [6,9]. Forensic field studies have measured host gene expression and microbial abundances to implicate the causes of mass mortality events in abalone and several other mollusc species as well as the causes of mortality of neonatal harbour seals [26–28]. Recently, the mystery of the near complete mortality of the sea urchin *Diadema antillarum* across the Caribbean in the 1980s was resolved using molecular techniques and fulfiling Koch's postulates in a recent outbreak, implicating a ciliate [29]. For SSW, a few studies have measured host gene expression in symptomatic and asymptomatic sea stars [30–32]. Fuess *et al.* measured gene expression of coelomocytes, an echinoderm immune cell, for three experimentally inoculated (sampled at late-stage SSW) versus three control sunflower stars, *Pycnopodia helianthoides*, to reveal differences in the expression of genes related to immunity, tissue remodelling and nervous system processes [30]. In the same species, comparing two asymptomatic and two wasting sea stars, Gudenkauf & Hewson [31] observed differences in the expression of cytoskeletal-associated proteins and matrix metalloproteinases, reflecting enhanced tissue degradation and matrix remodelling in symptomatic body wall tissues. In field samples of two late-stage wasting individuals and 13 asymptomatic *Pisaster ochraceus* sea stars, Ruiz-Ramos *et al.* [32] found differences in gene expression of toll-like innate immunity and apoptosis genes. However, the rapid progression of the disease from asymptomatic to signs of wasting to dead [13] has made it difficult to track changes through disease progression in the field. Thus, a longitudinal study of gene expression of more sea stars in controlled conditions would fill an important gap in our understanding.

Here we track changes in gene expression through time of 24 *Pisaster ochraceus* sea stars that remained apparently healthy/asymptomatic (8 individuals) or progressed through signs of SSW (16 individuals). All animals were asymptomatic in the field and on arrival to the laboratory; disease progression happened naturally without experimental infection as sea stars were maintained in individual aquaria in artificial seawater in the laboratory and sampled every 3 days for 15 days. In a previous study with the same animals, we identified changes in the host-related microbial community that were associated with the progression of signs of SSW [33]. In the present study, we generate host gene expression data and integrate with the previously generated microbiome data from the same tissue samples to provide insight into the role of microorganisms in, and reveal host responses to, disease. Our specific goals in this study were to (i) test for differences in host gene expression among apparently healthy, early- and late-stage wasting sea stars, with special attention to genes related to the echinoderm collagen system, immunity and hypoxia, (ii) test for associations between host gene expression and microbe growth rates/abundances with disease state, (iii) test if maintenance in artificial seawater in the laboratory affected the sea star microbiome and lastly, (iv) test for associations between genetic variants and whether animals wasted or remained asymptomatic.

## 2. Methods

### (a) Animal collection and experimental design

Thirty-eight adult, asymptomatic *P. ochraceus* (mean length from tip of ray to middle of the oral disc, $r = 9.7$ cm $\pm 2.2$ s.d.) were collected by hand by SCUBA from rocky subtidal reef 5–10 m depth around the wharf at Monterey Harbor, Monterey, California (36°36′21.44″N 121°53′23.69″W) in May and June 2016. Temperature at the time of collection was 12°C. Stars were shipped overnight to the University of Vermont with a total travel time of 17 h as previously described [33]. Briefly, sea stars were shipped in a styrofoam box with freezer packs and bunched up layers of newspaper to separate the freezer packs from the sea stars. Each star was individually packed in a plastic bag with a wet paper towel and approximately 300 ml of seawater. We selected Monterey as a collection site because it is in the middle portion of the species range, had been highly impacted by the disease in 2014, but was showing low signs of disease at the time of collection. We wanted to study exposed but apparently healthy animals to be able to study disease progression. Upon arrival to the laboratory, non-lethal biopsy punches were taken from the body wall (3.5 mm diameter biopsy punch, Robbins Instruments) and stars were photographed and examined for signs of SSWD. Biopsy punches were about the size of half of a grain of rice and only included epidermal tissue. Individuals were transferred to containers (31.5 cm × 18.5 cm × 11.5 cm) filled with 6.5 l of Instant Ocean artificial seawater (33 parts per thousand), each bubbled vigorously with individual bubblers, and kept in one of two incubators (SANYO MLR-350) at 12°C. Three times a week, containers were cleaned and refilled with Instant Ocean artificial seawater, then bubbled with the same bubbler to avoid cross-contamination. The stars were not fed during the experiment. Every 3 days for 15 days, non-lethal biopsy punches were taken from the epidermis of the body wall of each individual. If an individual was displaying wasting, epidermal tissue at the edge of a lesion was sampled. Given the specificity of the biopsies, individuals were classified by the 'symptom number' (see below) displayed at the time of sampling, regardless of signs of wasting the day before or after sampling. Biopsies were flash frozen in liquid nitrogen and stored at −80°C until RNA extraction. At each sampling time point, photographs were also taken and signs of wasting were recorded according to the *P. ochraceus* guide [34] with the inclusion of a fifth category for dead individuals. Symptom number classification was based on the Pacific Rocky Intertidal Monitoring Program 2017: (0) asymptomatic/apparently healthy (i.e. grossly normal with no signs of wasting); (1) one lesion on one ray or the central body; (2) lesions on two rays, one ray and the central body, or deteriorating rays; (3) lesions on most of the body and/or one or two missing rays; (4) severe tissue deterioration and/or three or more missing rays; with the additional classification of (5) dead. Individuals were classified as dead if there was absolutely no movement and turgor pressure had been lost from all tube feet. The experiment was terminated after 15 days due to the rapid progression of symptoms and the large number of biopsies collected with repeated sampling of individuals through time. Only 24 individuals were selected for the present study (all those that remained

asymptomatic and a random selection of those that wasted) due to the large number of biopsy samples with repeated sampling of individuals through time and the higher cost of eukaryotic RNA sequencing relative to 16S amplicon sequencing (see electronic supplementary material, table S1 for a schematic of individuals and time points used).

## (b) RNA extraction and sequencing library preparation

RNA was extracted from each biopsy punch using a modified TRIzol protocol (TRIzol reagent ThermoFisher Scientific). The tissue was lysed in 250 µl TRIzol with a plastic pestle then homogenized with 750 µl more TRIzol on a Vortex Genie2 (Scientific Industries) for 20 min. Two hundred microliters chloroform (ThermoFisher Scientific) was added to the mixture which was inverted 15 times, incubated for 3 min and spun for 15 min at $12\,000 \times g$ at 4°C. The aqueous phase containing RNA was transferred to a new tube and this step was repeated a second time. The RNA was precipitated from the aqueous phase by adding 500 µl isopropanol (ThermoFisher Scientific) and 1 µl 5 mg ml$^{-1}$ glycogen (Invitrogen), incubating for 10 min at room temperature and centrifuging for 5 min at $7500 \times g$ at 4°C. The RNA pellet was dried for 10 min at room temperature and resuspended in 50 µl nuclease-free water. The quality and quantity of the RNA extractions was measured using a NanoDrop 2000 Spectrophotometer (ThermoFisher Scientific) and Qubit 3.0 Fluorometer (Life Technologies). The RNA was checked for contaminating DNA by negative PCR amplification of the CO1 gene. RNA extractions from 93 samples were split in half and used to prepare total host mRNA libraries as well as 16S rRNA amplicon sequencing libraries using the Illumina TruSeq Stranded mRNA kit and the Illumina 16S metagenomic sequencing library preparation protocols (Illumina 2013) [33], respectively.

## (c) RNA sequencing, de novo transcriptome assembly and bacterial taxa table assembly

To increase sequencing depth per sample for host RNA, 93 samples taken from 24 of the original 38 individuals were chosen. Eight of these individuals ended the experiment apparently healthy and 16 showed signs of wasting. Through the course of the 15-day experiment, 54 of the 93 samples were taken from individuals that appeared grossly normal/asymptomatic (referred to as 'apparently healthy') and 39 were taken from individuals presenting with signs of wasting. See electronic supplementary material, table S1. Each library was uniquely barcoded and all libraries were sequenced across four Illumina HiSeq 3000 lanes ($2 \times 100$ bp). The quality of the reads was visualized with FastQC [35] and reads were trimmed and filtered with Trimmomatic [36] (see https://github.com/PespeniLab/ssw_rnaseq for data processing parameters). To generate a de novo transcriptome assembly for this species, we assembled reads from all samples from two individuals (one which was wasting-affected, number 38 and one which stayed asymptomatic, number 27) using the Trinity assembler [37]. We used samples from only two of the 24 individuals to minimize assembly errors due to increases in sequence heterozygosity with more individuals. We selected one asymptomatic and one wasting to maximize the chance of assembling transcripts that may only be expressed in one state and not the other. We mapped reads from all samples to this assembly using Bowtie2 [38]. Each read was allowed to multi-map in order to allow Corset [39] to group Trinity-assembled transcripts into longer length transcripts. We used Transdecoder to identify candidate protein-coding regions within the Trinity transcripts [40]. To further reduce noise in the transcriptome data and remove reads which were unlikely to have been sequenced from P. ochraceus RNA, we filtered transcripts that did not have a tblastx hit

(e value cutoff $1^{-5}$) to the transcriptome based on the published genome of Patiria miniata, another sea star species [41]. Annotations for transcripts in this final transcriptome ($n = 10\,004$) were based on a tblastx search to the NCBI Non-Redundant protein database limited to only echinoderms. This pipeline resulted in a counts matrix of the number of reads mapped to each transcript for each sample.

16S rRNA amplicon sequencing data was used to build an operational taxonomic units (OTUs) table as described previously [33]. Briefly, RNA from the extractions mentioned above was reverse transcribed to cDNA, which was used as template to amplify the V3-V4 region of the 16S gene. We chose to sequence from RNA (cDNA) rather than DNA because we were interested in detecting changes in metabolically active microbes and changes in growth rates of microbial taxa through time and across disease stages between samples rather numerical relative abundances; estimates of relative abundance from rDNA can be affected by relic DNA from dead taxa due to the stability of DNA [42,43]. 16S rRNA gene amplicon libraries were sequenced on the Illumina MiSeq platform using $2 \times 300$ base pair overlapping paired-end reads. Reads were clustered into representative OTUs using Qiime's Open Reference clustering strategy which uses a combination of database reference OTU clustering (using the Greengenes database) [44] and de novo OTU clustering, based on sequence similarity among reads. The result of this Qiime pipeline is a counts matrix, similar to the per transcript counts matrix mentioned above, which tallied read counts of each OTU for each sample.

## (d) Differential expression analysis

DESeq2 was used to correct for variation in library size and normalize the transcriptome counts matrix, and to test for differential expression among host genes [45] in R v.3.4.0 [46]. We ran models to test two hypotheses: (i) that there were overall differences in transcriptional patterns of tissue samples taken from wasting versus asymptomatic individuals; and (ii) that there were differences in gene expression between asymptomatic, early-stage and late-stage disease tissue samples. To test for differential expression between samples taken from wasting versus asymptomatic samples while controlling for the repeated measures of individuals sampled through time, we used the model ~individual + phenotype. To test for differential expression of genes between samples of different disease stages, we collapsed symptom classification groups 1 and 2 into 'early-stage disease' and 3, 4 and 5 into 'late-stage disease'. We then tested the difference between asymptomatic versus early-stage disease samples, asymptomatic versus late-stage disease samples, and early-stage disease versus late-stage disease samples with the model ~individual + disease_stage.

To test if gene categories with specific functional roles were enriched for differential expression, we performed rank-based gene ontology (GO) analyses with adaptive clustering using the Wald statistic outputs [47]. This rank-based test identifies GO terms enriched along a continuous distribution of a metric. In addition, given the rapid tissue disintegration and loss of rays in the progression of SSW, we were interested in identifying enrichment of transcripts related to collagen because of the importance of collagen in the adaptive detachment of a ray as a defensive mechanism and precursor to regeneration [48]. To do this, we developed a custom list of echinoderm-annotated, collagen-related genes based on blast annotations including the word 'collagen', which included the words 'collagen', 'collagenase' and 'collagenous'. We identified 51 transcripts annotated as being related to collagen; all but four of these genes were pro-collagen (i.e. encoding for different collagen isoforms or precursor molecules), while four were annotated as collagenases. We elected to use a custom list based on gene annotations to other

echinoderms rather than GO terms here because these annotations would be more accurate to the unique collagen systems of echinoderms. We then compared the Wald test statistic generated in the above DESeq2 models testing for differences in gene expression between wasting versus asymptomatic samples for collagen-related transcripts and the rest of the transcripts and tested for differences between these groups using a Mann–Whitney–Wilcoxon test. Similarly, we identified five genes as including the word 'hypoxia,' including the master regulator, hypoxia-inducible factor 1-α (HIF-1α) and other subunits of the transcriptional complex.

## (e) Host transcription and microbiome network analysis

The WGCNA package in R was used to (i) test for associations between changes in microbial growth/abundance with changes in gene expression across all samples, (ii) group the transcripts and OTUs into modules and (iii) correlate these modules with sample metadata [49]. For this analysis, we combined two counts matrices: one matrix containing the read counts of each transcript per sample and one matrix containing read counts of each OTU per sample. The transcript matrix contained 10 004 transcripts and the OTU matrix contained 1064 OTUs. The combined matrix was filtered to remove any genes or OTUs with zero counts in 25% of the samples. This left 3894 combined transcripts and OTUs: 3617 transcripts (36% of the total transcript matrix) and 277 OTUs (26% of the total OTU matrix). This combined, filtered matrix was normalized using the varianceStabilizingTransformation function in DESeq2 to account for differences in sequencing depth between libraries and then was quantile normalized to account for differences in distributions between the two matrices. Following normalization, the distributions of the transcripts and OTUs were not significantly different, which permitted integrating matrices to test for correlations of transcripts and OTUs.

WGCNA uses a soft threshold of the Pearson correlation between expression profiles of transcripts and OTUs to build networks. Here, we chose the soft thresholding power 4 based on the criterion of approximate scale-free topology. Modules were chosen with a minimum size of 40 genes or OTUs. Sample metadata used to test for correlations to modules included: day the sample was taken, the individual the sample was taken from, the phenotype of the individual (either asymptomatic or wasting) at the time the sample was taken, the phenotype number of the individual at the time of sampling according to the Pacific Rocky Intertidal *P. ochraceus* symptom guide (0–5), the disease stage of the individual at the time of sampling (asymptomatic, early-stage disease or late-stage disease), and the final phenotype at the end of the 15-day experiment of the individual the sample was taken from. Two samples (27-5-17 and 25-5-20) were identified as outliers based on hierarchical clustering and were removed.

## (f) Variant calling and outlier loci identification

Lastly, we identified single-nucleotide polymorphisms (SNPs) among the 24 sampled individuals and tested for associations between genetic variants and whether the star wasted or stayed asymptomatic. Reads from all samples were re-mapped to the final *de novo* assembly with Bowtie2 where reads were only allowed to map to one locus [38]. Samples were grouped by the identity of the individual, which resulted in increased sequencing coverage for each individual. We then used Samtools to merge all bam files, sort the compiled bam file, remove PCR duplicates, index and call variants [50]. A raw vcf file was made with bcftools and subsequently filtered with vcftools with the following parameters: four individuals were removed due to low-sequence coverage (individuals 7, 22, 24 and 26), only biallelic SNPs were kept, alleles with lower than 2% frequency were removed, loci with a mean depth less than 5 were removed, genotypes with a depth less than 4 were marked as

unknown, loci with a mapping quality less than 20 were removed, and finally loci with more than 10% missing data were removed. BayesScan2.1 was used to identify loci at different frequencies between individuals that wasted versus remained asymptomatic. Allele count data was extracted from the vcf file for input into BayeScan2.1 using make_bayescan_input.py from De Wit *et al.* [51]. BayeScan2.1 was run with default parameters [52].

# 3. Results

During the two-week experiment, eight individuals remained asymptomatic and 16 developed signs of wasting. Ninety-three biopsies were sequenced for total host mRNA (54 samples from individuals when asymptomatic and 39 samples from individuals when exhibiting wasting), resulting in 1.52 billion total reads (average 16.4 million per sample; 15.7 million after trimming for quality). The final *de novo* transcriptome assembly contained 10 004 transcripts with an average length of 954 base pairs per transcript, 8383 (83.8%) of which matched known echinoderm transcripts, 6498 (65.0%) of which had GO annotations and 220 (72.7%) BUSCO orthologues (electronic supplementary material, table S2).

## (a) Differential expression in epidermal tissue from asymptomatic versus wasting sea stars

Sixty-six transcripts were differentially expressed between samples taken from wasting versus asymptomatic individuals ($p_{adj} < 0.1$). Excluding samples taken from dead individuals, 33 transcripts were differentially expressed ($p_{adj} < 0.1$). Among transcripts with higher expression in asymptomatic than wasting stars were several immune-related genes (figure 1*a*). Among transcripts with higher expression in wasting than apparently asymptomatic stars were genes related to transcription and translation (figure 1*b*). Considering transcriptome-wide expression, there were 26 GO terms enriched for differential expression comparing asymptomatic versus wasting samples. Fifteen of the categories had higher expression in wasting relative to asymptomatic individuals and were largely related to RNA processing and regulation of gene expression (figure 1*c*). Eleven categories had higher expression in asymptomatic relative to wasting individuals and included functions related to cell adhesion, developmental processes and regulation of cytokine production (figure 1*c*). In addition, considering a custom annotated list of 51 collagen-related genes, there was significant enrichment for higher expression in asymptomatic versus wasting sea stars ($p < 0.0001$; figure 1*d*). Two other genes differentially expressed in this model included the transcription factor HIF-1-α ($p_{adj} < 0.001$; figure 2*a*) and collagenase 3, also known as matrix metalloproteinase-13 ($p_{adj} < 0.0001$; figure 2*b*), both with higher expression in wasting stars.

## (b) Differences in gene expression through sea star wasting progression

Comparing transcriptome-wide gene expression patterns in epidermal tissue among asymptomatic, early-stage and late-stage disease phenotypes, principal component analysis showed that variance among these groups was largely overlapping (figure 3*a*). However, when testing for differences in gene expression between groups, 17 transcripts were

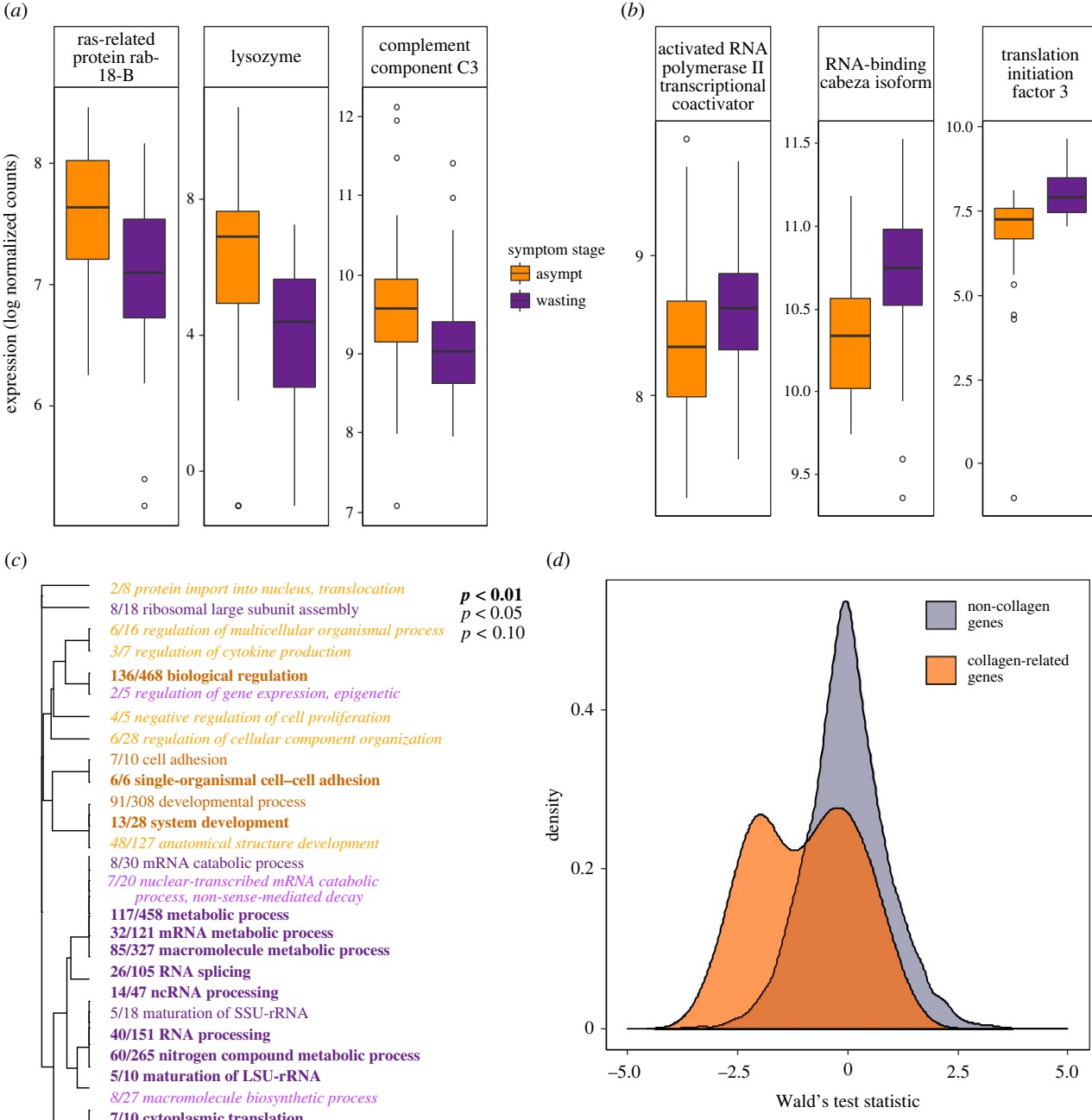

**Figure 1.** Differences in gene expression between asymptomatic and wasting sea stars in common garden conditions. Log-normalized expression of three immune-related genes upregulated in asymptomatic samples relative to wasting (*a*) and three transcription and translation-related genes upregulated in wasting samples relative to asymptomatic (*b*). GO categories enriched by genes upregulated (orange) or downregulated (purple) in asymptomatic individuals compared to wasting individuals, summarized by biological process (*c*). The fraction preceding the GO term indicates the number of genes annotated with the term that pass *p*-value threshold of 0.05. (*d*) Density plots of the Wald's test statistic of each gene in the differential expression analysis comparing wasting to asymptomatic animals. Transcripts with annotations related to collagen (orange) are separated from all other transcripts (grey). Negative test statistic values indicate lower gene expression in samples from wasting relative to asymptomatic individuals.

differentially expressed between asymptomatic and early-stage samples with 19 GO terms enriched considering expression differences across all transcripts. Twelve of these GO terms were enriched for higher gene expression in tissue samples from early-stage disease compared to asymptomatic individuals and included terms related to RNA processing, macromolecule metabolic processes and ncRNA processing (figure 3*b*, green). Seven GO terms were enriched for higher gene expression in tissue samples from asymptomatic individuals compared to early-stage disease individuals and included terms related to developmental processes, anatomical structure development and cell–cell

adhesion (figure 3*b*, orange). Like the asymptomatic versus wasting contrast above, there was enrichment for higher expression of the collagen-related genes, primarily collagen genesis genes, in samples taken from asymptomatic individuals compared to early-stage disease individuals ($p < 0.0001$). Comparing early- and late-stage disease samples, 77 transcripts were differentially expressed with 1 GO term enriched: organonitrogen compound catabolic processes. Sixty-eight of the differentially expressed transcripts had higher expression in early-stage disease samples relative to late and included some genes related to collagen production, though there was not enrichment in collagen-related genes in

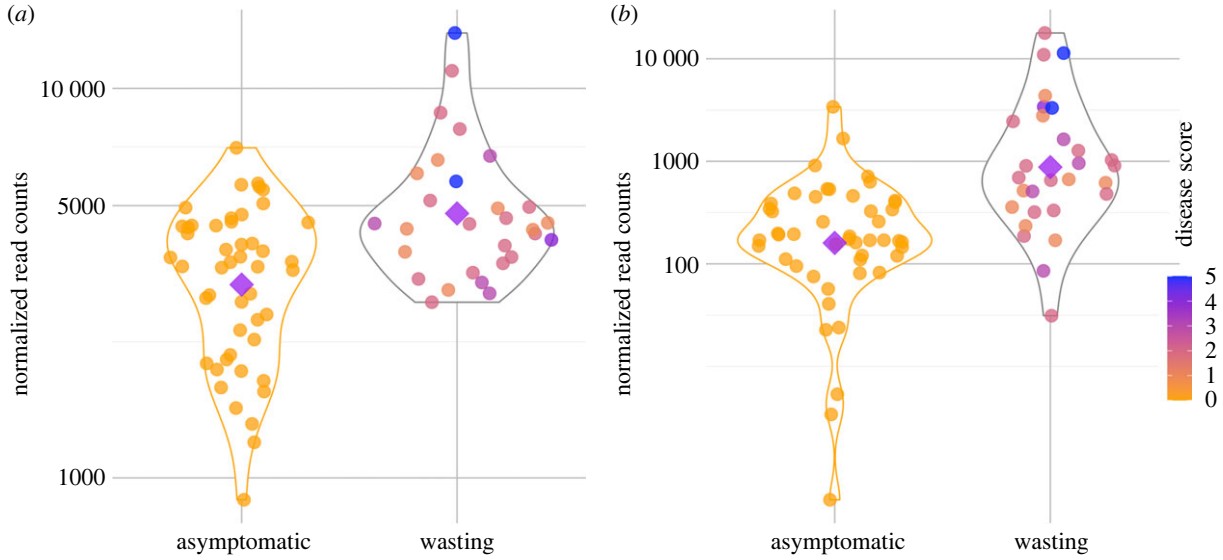

**Figure 2.** Differences in gene expression of (a) HIF-1-α ($p_{adj} < 0.001$) and (b) collagenase 3, matrix metalloproteinase-13 ($p_{adj} < 0.0001$) for asymptomatic and wasting sea stars. Each point represents expression from a biopsy.

this model ($p = 0.08$). Lastly, comparing asymptomatic versus late-stage disease samples, 248 transcripts were differentially expressed: 146 were upregulated in asymptomatic samples and 102 were upregulated in late-stage disease samples, including a transcript annotated as the toll-like receptor 2. Thirty-five GO terms were enriched for this model; 24 were upregulated in asymptomatic samples and included regulation of immune effector process, regulation of lymphocyte-mediated immunity, anatomical structure morphogenesis and development and cell adhesion (figure 3c, orange); 11 were upregulated in late-stage disease samples and included RNA processing, ncRNA processing, macromolecule metabolic process and translation (figure 3c, blue).

## (c) Correlations between host transcription and microbe abundance

Combined gene expression and OTU abundance data clustered into eight distinct modules (electronic supplementary material, figure S1). Five modules were significantly correlated with one or more sample metadata category (electronic supplementary material, figure S1). Four of these modules contained both transcripts and OTUs, representing networks where transcript and OTU abundances were correlated. Transcripts and OTUs in the black module (31 transcripts and 86 OTUs) were significantly correlated with day, phenotype, phenotype number and disease stage (figure 4). Overall, growth rates/abundances of OTUs in the black module were lower in asymptomatic samples compared to early- or late-disease stage samples whereas the expression of transcripts in the module were higher in asymptomatic samples compared to early- or late-stage samples (figure 4). The genes in this network were related to cell adhesion, host defense and collagen proteolysis (among other processes). Specifically, genes include fibronectin and cathepsin L, a glycoprotein and protease, respectively, involved in the formation and degradation of extracellular matrix components. The OTUs in this network included *Pseudoalteromonas* spp. and *Polaribacter* spp. taxa that were

previously identified as differentially abundant between individuals that wasted and those that stayed asymptomatic [33].

## (d) Stability of the microbiome in artificial seawater

To test if containment in laboratory conditions, individual aquaria with artificial seawater, affected the microbial assemblages of the sea star epidermis, we tested for changes in microbiome composition through time for the eight sea stars that remained asymptomatic through the duration of the experiment, day 0 though day 15. Day 0 biopsies were on arrival after overnight shipping and before introduction to artificial seawater and thus represent the natural microbiome composition. We found that microbial diversity was relatively stable through time with no significant differences in α diversity pairwise between days except for between days 0 and 6 and days 6 and 9 (Faith's diversity, pairwise Kruskal–Wallis, $q < 0.05$), suggesting an initial shift in the microbial community between days 0 and 6, but a return to day 0 diversity levels by day 9. Taxonomic composition patterns (β-diversity) mirrored this result (figure 5a), showing a shift by day 6 and a return by day 9 with differences in relative abundance driven by changes in lower abundance taxa. Weighted and unweighted UniFrac tests for differences in community composition (β-diversity) revealed a similar pattern with no differences when considering weighted dissimilarity (PERMANOVA pseudo-$F = 0.938$, $p = 0.489$) and differences when considering unweighted dissimilarity (PERMANOVA pseudo-$F = 2.081$, $p = 0.001$), again showing the differences were driven by rare or low abundance taxa (figure 5b). Taken together, these results suggest that overall, there was little impact of laboratory conditions on microbial community composition.

## (e) Single-nucleotide polymorphisms related to final disease phenotype

After filtering, we identified 98 145 high-quality SNPs. Comparing seven individuals which remained apparently healthy and 13 that showed signs of wasting, we did not identify any

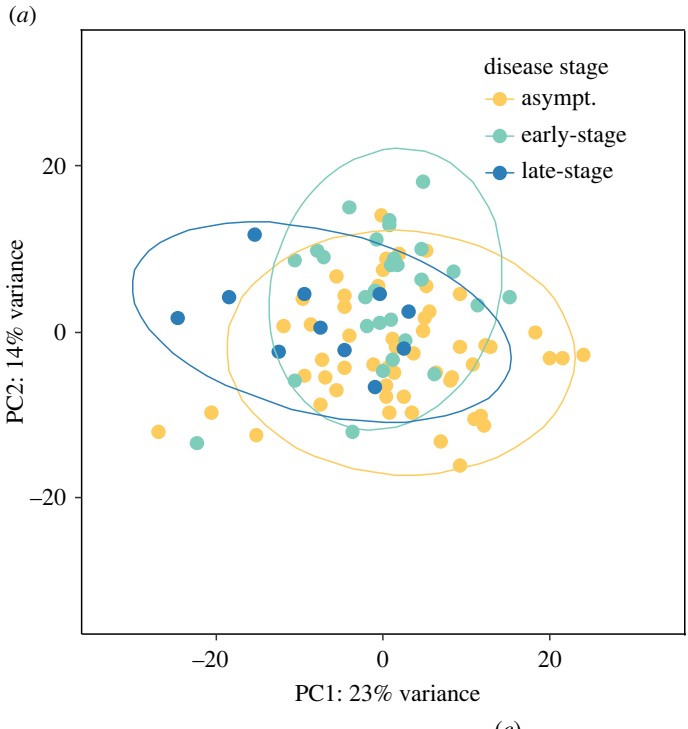

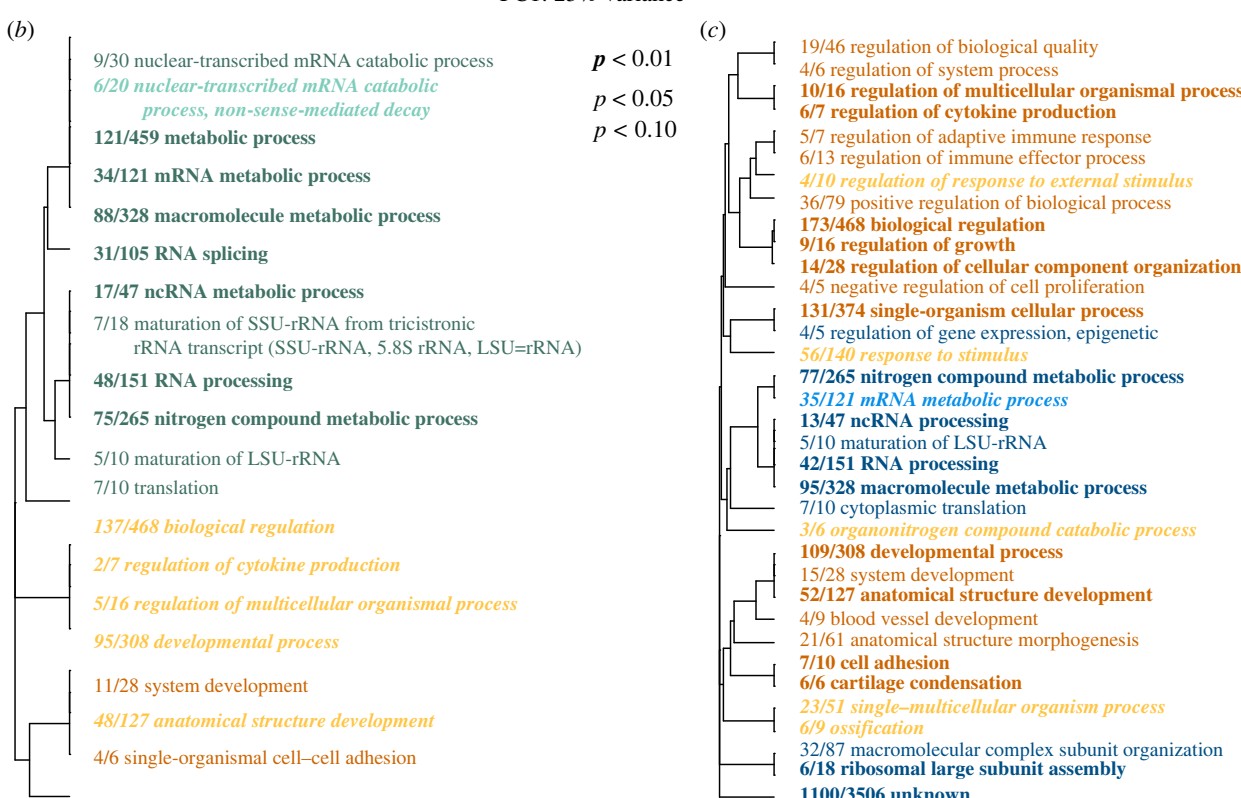

**Figure 3.** Differences in gene expression between asymptomatic, early-stage and late-stage wasting individuals. (*a*) Principal component analysis plot of transcript expression in samples taken from asymptomatic individuals (orange), early-stage disease animals (teal) or late-stage disease animals (blue). Ellipses are drawn around each group's centroid (confidence interval, 0.95). (*b*) Biological process GO categories enriched by genes upregulated (orange) or downregulated (green) in asymptomatic versus early-stage wasting individuals. The size and boldness of the font indicates the significance of the term as indicated by the legend. The fraction preceding the GO term indicates the number of genes annotated with the term that pass an unadjusted *p*-value threshold of 0.05. (*c*) Biological process GO categories enriched by genes upregulated (orange) or downregulated (blue) in asymptomatic versus late-stage wasting individuals.

loci associated with final health status, suggesting that resistance to wasting does not have a genetic basis.

## 4. Discussion

We measured gene expression longitudinally for 24 *Pisaster ochraceus* sea stars as they remained asymptomatic or wasted over a 15-day period. We find that visibly healthy individuals had higher expression of immune-related, pro-collagen and cell–cell adhesion genes, relative to early- and late-stage wasting stars. Specifically, (i) we find a relatively muted immune response in wasting stars with a relatively higher expression of immune genes in animals that stay asymptomatic. We find strong evidence that sea stars experiencing wasting disease have lost control of their mutable

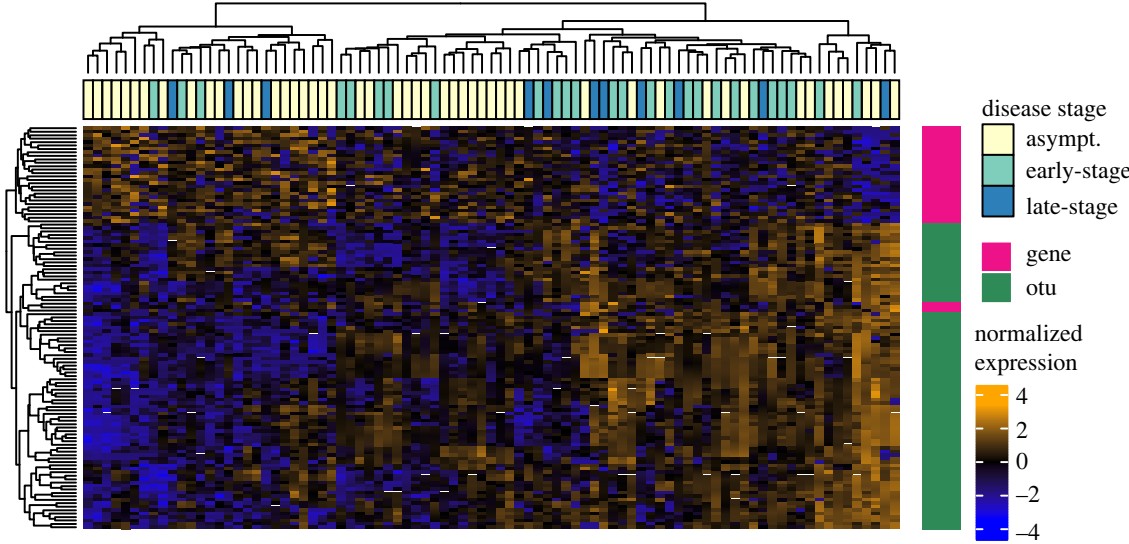

**Figure 4.** Growth/abundances of correlated transcripts and microbes. Heatmap of normalized expression/abundance of all transcripts and OTUs per sample in the black module identified by WGCNA. Rows are annotated pink for genes or green for OTUs. Columns are annotated yellow for asymptomatic, teal for early-stage wasting or blue for late-stage wasting individuals.

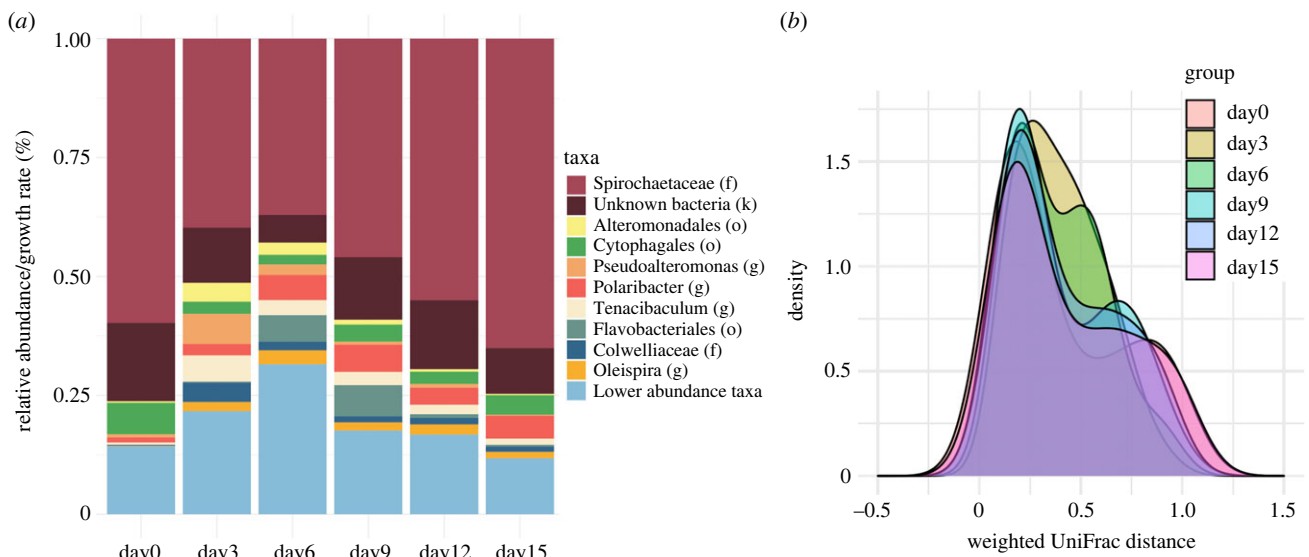

**Figure 5.** Resilience and stability of the microbiome of asymptomatic sea stars through time in laboratory conditions. (a) Time course of microbiome composition of the eight sea stars that remained visibly asymptomatic through the duration of the experiment; (b) minimal changes in community composition in pairwise comparisons across days (weighted Unifrac PERMANOVA pseudo-$F = 0.938$, $p = 0.489$).

collagenous tissue system showing reduced and variable expression of collagen, cell adhesion and tissue remodelling-related genes. We find evidence that epidermal cells of wasting stars are experiencing hypoxia with the upregulation of HIF-1, a key regulator of apoptosis during hypoxia [53]. (ii) Using network analyses to integrate host gene expression with microbiome data from the same tissue biopsies, we find evidence for host–microbe interactions as host transcripts and microbe growth/abundances showed correlated changes across samples in association with health status. As above, the genes in these modules perform functions related to immunity, collagen processes and cell adhesion, while the microbes were among those previously identified as associate with health status in SSW [33]. (iii) Considering animals that remained asymptomatic across the course of the study, we find stability and resilience in microbial community composition despite laboratory conditions

including artificial seawater, in contrast with other laboratory studies of coral and horseshoe crabs [54,55], but consistent with the observations of a core microbiome for *Pisaster ochraceus* [56] and *Pycnopodia helianthoides* [57] in the field. Interestingly, all four studies, including the present study, find Spirochaetes to be the dominant microbes of asymptomatic stars for both *P. ochraceus* and *P. helianthoides* and across multiple geographical locations and tissues [33,56,57]. The microbiome resilience we observe through time in the laboratory could be related to resistance to SSW, though further investigations are needed. Lastly, (iv) we find no evidence of genetic variants that associate with susceptibility to SSW, similar to Burton *et al.* [58], but in contrast with Schiebelhut *et al.* [59]. These results expand our understanding of how sea stars react to and potentially resist the cause(s) of wasting and implicate environmental and microbial drivers of disease.

Tracking host gene expression changes through the course of disease progression, we found the overrepresentation of genes related to cell adhesion and immunity upregulated in asymptomatic compared to wasting samples. By contrast, in the transition from asymptomatic to early-stage disease, wasting animals showed higher expression of genes related to RNA processing, transcription and translation. As early-stage wasting individuals transitioned to late-stages of disease, genes related to translation, hypoxia response and apoptosis were upregulated while genes related to collagen production were further downregulated. Taken together, these longitudinal gene expression patterns suggest that asymptomatic animals remained asymptomatic using an active immune response and that both early- and late-stage wasting animals lost the regulation of systems that maintain tissue integrity. Though tracking changes in gene expression and microbiome composition of individual sea stars in the wild is unfeasible, it is important to note that these results may be specific to our laboratory conditions and may not represent the changes occurring during wasting in the wild. However, a study comparing the microbiomes of field-sampled naive, exposed and wasting sunflower sea stars, *Pycnopodia helianthoides*, in lieu of longitudinal sampling of individuals, in Alaska when the panzootic first reached the area, reveals a proliferation of anaerobic microbes with exposure and further with wasting [57], further supporting the idea that low oxygen may play an important role in SSW [19].

We observed a general pattern of higher expression of immune-related genes in asymptomatic relative to wasting sea stars, in contrast with what other studies of SSW have found [30–32]. *P. ochraceus* individuals used in the present study were collected from Monterey Bay, CA, an area that had previously experienced SSW. Most, if not all, of the individuals in this experiment likely had been exposed to the causes of SSW. Individuals that remained asymptomatic throughout the experiment were perhaps able to mount an appropriate immune response using the advanced innate immune system of echinoderms [60] to avoid developing signs of SSW. Our results support this hypothesis. Immune genes such as Ras-related protein Rab-18, lysozymes, and Complement C3 were at relatively higher transcription levels in asymptomatic samples compared to wasting samples. All of these genes have been identified previously in echinoderms as being upregulated in response to bacterial lipopolysaccharides [61,62].

Results of this study also show a strong signature of upregulation of genes related to tissue integrity in asymptomatic versus wasting individuals. These genes are of particular interest as they relate to the dramatic progression of SSW, from lesions to tissue melting away and loss of rays. We found enrichment in genes related to cell adhesion, structure development and collagen upregulated in asymptomatic versus wasting samples in all contrasts. Similar genes related to collagen and tissue remodelling were differentially expressed in previous studies of SSW [30,31]. Collagen in echinoderms can change between a stiff and pliable consistency very quickly, switching from a very strong holding capacity to a soft consistency [63]. This change in collagen consistency precedes the adaptive detachment of a ray as a defensive mechanism and as a precursor to regeneration [64]. Individuals that remained asymptomatic in this experiment were likely able to retain control over this neurologically controlled process. Potentially related,

wasting relative to asymptomatic sea stars had a strong signature of upregulation of RNA processing genes, which is associated with many neurological diseases in humans [65]. If upregulation of RNA processing functions is evidence of neurological disease or dysfunction in sea stars, wasting stars could lose control of their neurologically controlled catch collagen system resulting in tissue disintegration.

We identify correlated transcripts and microbes whose abundances/growth rates relate to sea star health status. In one of the networks, the increased abundance of microbes through disease stages was correlated to decreased expression of host genes (figure 4). This could represent direct interactions between host and microbiome where the microbiome is affecting host transcription or host transcription is affecting the abundance of microbes in the microbiome, as has been observed in human conditions [66–68]. Transcripts in this module perform functions related to adhesion, including fibronectin, IgGFc-binding protein and cathepsin (among others) [69,70]. The group contained OTUs belonging the *Pseudoalteromoas* genus, which contains taxa that produce antibiotic and antimicrobial compounds [71], as well as the genus *Tenacibaculum* which contains marine pathogens [72]. *Polaribacter* spp., among other copiotrophic microbes that thrive in carbon-rich, low-oxygen environments (*Flavobacteriaceae* and *Rhodobacteriacaea*) [19,25,33], was also found in this network and showed increased in abundance through disease progression (figure 4c). The patterns of growth/abundance of these taxa paired with the upregulation of the master hypoxia regulator, HIF-1α, and the downregulation of pro-collagen systems with wasting, suggest a relationship between microbial dysbiosis and the decay of sea star tissue. The decay of sea star tissue could further reduce oxygen conditions and support the proliferation of copiotrophic microbes [19]. However, complete metagenomic sequencing and functional studies are needed to identify the specific microbial taxa and the genes in their genomes that may be eliciting a reaction in the host.

Taken together, the shift in microbiome with SSW and the host response to hypoxia, suggest interacting biotic and abiotic factors could drive disease. This could be similar to other mass mortality events that seem to plague echinoderms. For example, a pathogen decimated populations of the long-spined sea urchin, *Diadema antillarum* (93–98% mortality), sweeping across the Caribbean in 1 year, 1983 [73]. The cause has been recently attributed to an ciliate [29], but the conditions that trigger the mass proliferation of the ciliate are unresolved. More recently, with a similar set of symptoms to SSW, skin ulceration disease (SKUD) has affected multiple species of sea cucumber across the world [74]. Interestingly, a range of factors can induce SKUD, including isolated pathogens, injection or ingestion of organic matter and decreased temperatures [74–76], suggesting that the disease may be environmentally cued autocatalytic stress responses. Strikingly, similar symptoms to SSW and SKUD can be induced by injecting media that promotes the growth of *Vibrio* spp. in the crown of thorns sea star, *Acanthaster planci* [77]. In addition, media-induced disease can be transmitted by contact between crown-of-thorns and another sea star species, *Linckia guildingi* [78]. A pattern is emerging across echinoderm diseases: factors that promote the proliferation of one or several microbes could lead to dysbiosis, loss of neurological control of the echinoderm catch collagen system, followed by lesions and loss of tissue integrity. The decaying tissue

could then lead to hypoxia and death. However, further field, manipulative and longitudinal studies are needed.

## 5. Conclusion

Our results suggest that sea stars that stay apparently healthy while others collected from the same environment waste may do so through an ability to launch an immune response and maintain tissue integrity through sustained control of collagen and cell adhesion functions. Integrating with microbiome data from the same tissues results suggest a compromised immune system, loss of control of the collagen system, and the proliferation of specific microbes could contribute to the progression of SSW disease. Integrative genomic studies as well as increased monitoring of environmental conditions in the marine environment [6,79], including oxygen levels in intertidal and subtidal habitats and tracking of runoff from terrestrial environments, phytoplankton blooms, temperature anomalies and disease monitoring, will improve our understanding of the interacting factors driving disease outbreaks in the oceans.

Data accessibility. The data from this study are available from the Dryad Digital Repository [80]. Sequence data reported in this article have been deposited in the National Center for Biotechnology Information (NCBI) (BioProject no. PRJNA934423).

Additional information is provided in the electronic supplementary material [81].

Authors' contributions. M.H.P.: conceptualization, formal analysis, funding acquisition, project administration, visualization, writing—original draft and writing—review and editing; M.M.L.: conceptualization, data curation, formal analysis, funding acquisition, investigation, methodology, validation, visualization and writing—original draft.

All authors gave final approval for publication and agreed to be held accountable for the work performed therein.

Conflict of interest declaration. The authors declare no competing interests.

Funding. This work was supported by the National Science Foundation RAPID grant no. IOS-1555058 (to M.H.P.).

Acknowledgements. We thank B. Nesnevich and H. Hargarten for assistance with animal care, and T. Fay for assistance with collections. We also thank M. Austin, UVM's Enterprise Technology Services and the Vermont Advanced Computing Core for computing support. We are also grateful to the Pespeni lab members, particularly A. McCracken, C. Petak and A. Hall for thoughtful, constructive feedback on the manuscript.

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
