## [Peer Review File · Proceedings of the Royal Society B: Biological Sciences]

Review History

RSPB-2021-0772.R0 (Original submission)

Review form: Reviewer 1

Recommendation

Major revision is needed (please make suggestions in comments)

Scientific importance: Is the manuscript an original and important contribution to its field?

Excellent

General interest: Is the paper of sufficient general interest?

Good

Quality of the paper: Is the overall quality of the paper suitable?

Marginal

Is the length of the paper justified?

Yes

Should the paper be seen by a specialist statistical reviewer?

No

Do you have any concerns about statistical analyses in this paper? If so, please specify them explicitly in your report.

No

It is a condition of publication that authors make their supporting data, code and materials available - either as supplementary material or hosted in an external repository. Please rate, if applicable, the supporting data on the following criteria.

Is it accessible?

Yes

Is it clear?

Yes

Is it adequate?

Yes

Do you have any ethical concerns with this paper?

No

Comments to the Author

General Comments

Sea star wasting disease has gained prominence in recent years due to its widespread mortality on the North American Pacific Coast. The etiology of the condition is not fully resolved, and many environmental or microbial insults result in similar disease signs, but no single factor can consistently explain wasting outbreaks. Lloyd and Pespeni extend on a previous study (Lloyd & Pespeni, 2018 Sci. Rep), which surveyed microbiome composition during wasting progression, to look at host gene expression, focusing especially on genes involved in collagen and hypoxia response which have recently been suggested as disease triggers. They found that the expression of genes involved in collagen (presumably synthesis) was greater in specimens that did not waste during their 18 d experiment than in specimens that wasted, and found evidence that hypoxia-response (notably HIF1a) had greater expression in animals that did waste. They extend transcriptome information to previously sequenced 16S rRNAs to identify linkage between these components and conclude that greater abundance of 'beneficial' microorganisms in healthy specimens transitioned to more 'pathogenic' and 'copiotrophic' taxa as individuals wasted.

There is really useful and interesting data on host gene transcription buried in this manuscript – the author's finding of collagen- and hypoxia-response gene patterns echo recent work showing that low oxygen conditions generated at the animal's surface may relate to wasting response (Aquino et al. 2021), and this falls in line with work showing that wasting is a basal-to-surface process affecting collagen and body wall tissues (see: DOI: <https://doi.org/10.3354/dao03598>). However, this reviewer has significant concerns about the interpretation of the microbiome data presented in this manuscript (and originally presented in the Lloyd and Pespeni 2018 Sci Rep paper). The biggest concerns regarding the microbiome part of this manuscript are:

A. The authors retrieved sea stars from Monterey, California, shipped them to Vermont, and then maintained them in artificial seawater (ASW; Instant Ocean). While ASW can be used to mimic the inorganic constituents of seawater quite well – and metazoans can be maintained in captivity using this medium – it does not mimic naturally occurring organic compounds in seawater. Marine bacteria are highly sensitive to containment (bottle effects); the surface microbiome of captive sea stars likely mostly represents taxa that consume mucus and other substrates put out by the sea stars themselves – rather than those that are firmly embedded in

tissues. There is every expectation that these will change rapidly if sea stars are transferred into ASW (see Kooperman et al., 2007 FEMS Microb Lett; Friel et al., 2020 Front Micro). This reviewer disagrees that host-associated microbiomes, especially surface-associated microorganisms can be studied using samples maintained in ASW for this reason. Host gene transcription, on the other hand, is likely less affected by ASW conditions.

B. The authors analyzed microbiome data by performing RT-PCR on RNA extracts, rather than PCR of DNA extracts. These answer two different questions. rRNAs (c.f. rDNA) levels vary between taxa primarily by their relative activity but also growth strategies (see Steven et al., 2017 Applied and Environmental Microbiology; Bowsher et al., 2019 mSystems). It is possible, for example, for numerically rare taxa to be fast growing and produce lots of rRNA, but numerically abundant taxa to be slower-growing and produce fewer rRNAs (see Gremoin et al., 2003 Environmental Microbiology). In a community sense, rRNA tells you the relative activity of different constituents and rDNA tells you (with PCR bias caveats) numerical relative abundance. For this reason, rRNA-based approaches cannot provide relative abundance data for microbial communities.

C. The authors never qualify satisfactorily trait-based assignments to 16S rRNA matches. Just because a bacterial genus has pathogenic or “beneficial” representative does not mean that it is possible to say that these genera all share that trait – in fact, for most bacterial genera, pathogens are the exception to the rule for genus-level assignments. I’m still not sure how “known beneficial” could be derived from previous work, unless there were co-culture based studies which illustrated enhanced metazoan growth rates or immune function, etc. An alternate approach to labeling bacterial taxa with these traits is to focus on their relative growth capacity based on genome comparisons – see for example Haggerty & Dinsdale, 2016 Global Ecol & Biogeog. Unless taxonomic assignments are made at 100% similarity across the 16S rRNA, this reviewer does not believe assignment to “pathogen” is possible – and even then, microorganisms that are 100% the same across this gene can have vastly different gene inventories (see Jaspers & Overmann, 2004, Appl Env Micro).

On the other hand, the data on host gene transcription is interesting and, in this reviewer’s opinion, meriting publication on its own, either by removing the microbiome data altogether, or reducing it and focusing on general growth strategies of bacterial genera – and acknowledging limitation of aquarium-based studies and their impacts on microbiome constituents. I’ve made many suggestions below which I hope assist the authors in a revision.

Line 11: I recommend replacing “healthy” and “sick” with their veterinary terms “grossly normal” (or “not wasting”) and “wasting-affected” here and throughout the manuscript

Line 12: Were the “collagen” genes actual genes involved in collagen generation, or were these collagenases or remodeling genes? It would probably be best to go with a gene functional ontology which is consistent with other studies.

Line 13: Same comment here: “hypoxia-related” genes – are these response to hypoxia or caused by hypoxia or...? A little later in the manuscript I read that these include “hypoxia inducible factor” – if this is the only hypoxia-related gene I would recommend keeping it as “hypoxia inducible factor”

Line 16: The authors list here and throughout the manuscript three “known” traits for genera, including “beneficial”, “pathogenic” and “copiotrophic”. It’s unclear how or who established these traits for all species within these genera – i.e. how are these “known”?

Line 17-18: “... considering genotypes of stars at 98,145 SNPs, no variants...” this doesn’t make sense. Perhaps replace “at” with “had” as appropriate

Lines 24 – 35: While I think this is well put and a nice summary of infectious diseases, it sets up the manuscript as an investigation of an infectious disease. There is currently no evidence that this is the case for sea star wasting disease, or any echinoderm-related disease as well. Rather,

these may be environmentally-cued autocatalytic stress responses. See for example Delroisse et al., 2020 Sci Rep and sea cucumber skin ulceration disease.

Lines 67 – 69: Again, this statement implicates that there is a pathogen involved in sea star wasting – it might be better to state this as “could provide insight into the role of microorganisms in, and reveal host responses to, disease”

Line 71 – 73: See above comment about the limitations of aquarium-based studies. I would also caution against the identification of “healthy” microbiomes – rather, these are microbiomes that are observed before wasting occurs.

Line 87 – 88: Were the specimens shipped in water? On ice?

Lines 93 – 94: Instant ocean is a mix of inorganic salts and other ions that mimics the chemical environment of seawater, but as mentioned above does not mimic organic constituents of seawater on which heterotrophic bacteria rely – for example, coral reef aquariums are typically amended with carbon sources since they are carbon limited. Bacterial/archaeal communities inhabiting aquariums are very different from natural environments (unless they are symbiotic/embedded within tissues). Most bacteria in biopsy punches are probably at the animal’s surface, which will change greatly in captivity

Line 96: Biopsy punches were taken every 3 days for 15 days over the course of the experiment. How do these line up with the appearance of wasting? For example, if wasting occurred half-way through these 3 days, then which biopsy was used? This raises an important consideration when comparing early vs late wasting – if wasting is observed a day after a punch, and then considered early wasting, and in another star wasting starts on the day that a punch is taken and also considered early wasting, both gene expression and microbiome constituents reflect different stages of disease progression.

Line 121: Please include “16S rRNA amplicon” – not just “16S amplicon”. See above for comment on basing quantitative microbiome interpretations on RNA- (vs. DNA-)template libraries.

Line 127: A conceptual diagram showing when animals became sick and when transcriptomic/16S rRNA amplicon library samples were collected would be helpful. Did all “sick” specimens do so at the same time? Line 128: So all analyses were performed on singular specimens? (“from a healthy individual”).

Line 153 – Line 155: If I understand this analysis correctly, this would seemingly be a quantitative interpretation of bacterial OTUs. Because of the problems in interpretation of RNA-based libraries, it would be more appropriate to use non-quantitative statistics on their presence/absence (of ‘active’ bacteria).

Line 160 – 161: “at first onset of SSWD” – how was this captured when biopsy punches were taken every 3 days?

Line 166: “immediately before and after” – so this means that onset occurred within these 3 day windows?

Lines 191 – 204: See above – the authors should explore transcript read count vs OTU presence/absence.

Line 240: 6×10^4 to 4×10^7 is a huge range of variation of reads per sample. Was any rarefaction performed to normalize edge effects?

Lines 251 – 254: The presentation of broad categories (ontologies?) in the text is discordant with Figure 1. Figure 1a in the copy I review shows three immune-related genes; Figure 1b shows

three transcription/translated related genes. The text mentions structural related genes, and it's unclear which of the ontologies in this figure correspond with this category. Likewise, the figure identifies collagen-genes, but I can't find what genes these actually are in either the figure or elsewhere in the text.

Line 264-265 and Fig. 2: So the only hypoxia related gene is HIF 1a – so it would be better to state this throughout the manuscript instead of “hypoxia-related genes” (plural). Was collagenase 3 the only collagen-related gene? I think “collagen-degradation and remodeling gene” would be a more accurate moniker.

Line 301: The sentence beginning with “While” should perhaps start with “Seven”.

Line 305: So if the “collagen-related” genes were collagenases, and they were more expressed in healthy c.f. wasting stars, this would mean that collagen is being broken down, potentially, more in healthy stars than wasting stars. If there are more genes within this ontology – for example, collagen genesis, then I'd recommend splitting these out to reflect constructive and destructive processes.

Line 310: “genes related to collagen production” – ah, so this includes both production and destructive processes. I would recommend splitting these. Are these genes all echinoderm transcripts or might these include bacterial transcripts as well? Even with polyA selection, there is always some contamination with prokaryotic transcripts within transcriptomes.

Lines 308 – 316: The comparison between healthy and late-stage disease that shows an upregulation of immune genes is not surprising, since at this stage the specimens are infiltrated by bacteria and other microorganisms through lesions.

What would really help in this list of up- and down-regulated transcripts could be a conceptual figure showing a timeline of disease progression with the various categories.

Line 331: While it is certainly interesting to see trends in microbiome composition and correlations to host transcripts, there could well be correlation between un-related processes. Add to this the difficulty in quantitative interpretation based on RNA libraries, and it is this reviewer's opinion that this part of the manuscript does not add much to knowledge of disease process.

Line 358: The last sentence of this paragraph is not complete

Line 368 – 369: Sea star wasting disease has never been shown to be highly transmissible.

Line 372 – 375: Decreases in relative abundance during disease progression could relate to increases in absolute abundance of other groups – i.e. they may just grow more slowly. It is not appropriate in this reviewer's mind to attribute these as “healthy” or “beneficial” microbiome constituents without rigorous testing to compare health states with and without these organisms. At the very least, quantitative (vs relative i.e. amplicon libraries) approaches are necessary to examine their dynamics during disease progression before these can be identified with any trait.

Line 385 – 387: It is not exactly a hypothesis that there are some microbiome constituents correlate with patterns of gene expression in the host.

Lines 400 – 410: Much of this is a restatement of the results. It would be more appropriate to focus on the authors interpretation (i.e. lines 408 – 410), and place these at the start.

Lines 434 – 450: Most of this section is speculation. Sure, there are species within the genera identified that can produce e.g. antibiotic and antimicrobial compounds, or may be pathogenic, but extending this to the broad number of taxa within a single genus is highly tenuous.

Line 454: Unfortunately, the evidence for a healthy microbiome implicated in constantly healthy specimens is not warranted by the data provided (alone).

Review form: Reviewer 2

Recommendation

Reject - article is scientifically unsound

Scientific importance: Is the manuscript an original and important contribution to its field?

Marginal

General interest: Is the paper of sufficient general interest?

Marginal

Quality of the paper: Is the overall quality of the paper suitable?

Marginal

Is the length of the paper justified?

Yes

Should the paper be seen by a specialist statistical reviewer?

No

Do you have any concerns about statistical analyses in this paper? If so, please specify them explicitly in your report.

No

It is a condition of publication that authors make their supporting data, code and materials available - either as supplementary material or hosted in an external repository. Please rate, if applicable, the supporting data on the following criteria.

Is it accessible?

Yes

Is it clear?

Yes

Is it adequate?

Yes

Do you have any ethical concerns with this paper?

No

Comments to the Author

Thirty eight asymptomatic *Pisaster ochraceus* seastars were flown from Monterey California to Burlington Vt and held in 1 gallon containers w artificial seawater (3 water changes/week). IN a 15 day lab observation, 16 stars became sick and 8 remained healthy. What happened with the other 14 that were shipped? Were they dead or sick on arrival, suggesting the transit was stressful? IN the Results, line 1 it says: During the two-week experiment, 29 individuals showed signs of SSWD and 8 remained healthy.

The surface microbiome and immune related genes were monitored in 24 of the stars for 15 days. Why was the experiment terminated at 15 days? Because more and more stars were becoming

sick and dying? Inferences were made about causative agent and immune function based on comparing whether the outcome was to become sick or remain healthy. Caution is needed in interpreting microbiome trajectories since this is a lab experiment with long-distance shipped animals that might already be stressed by transport and lab conditions. Further caution is needed in inferring cause and effect between immune condition and bacterial surface changes. While the authors conclude that a healthy immune response drives a beneficial microbiome, its also possible that a healthy microbiome is permissive of a healthy immune response.

The study concludes that “Animals that remained healthy had an active immune response and a beneficial microbial community while animals that became sick showed evidence of responding to hypoxia and a proliferation of opportunistic microbes that thrive in carbon-rich, oxygen-poor environments.”

The immune data are the most useful. The authors found higher expression of immune-related, tissue integrity and collagen genes in healthy relative to sick individuals. I think the result of lower expression of collagen-related genes in sick relative to healthy is interesting and deserves further investigation. The finding of higher expression of hypoxia-related genes in sick relative to healthy individuals is harder to interpret, since this could be an artifact of lab stress conditions.

I find the microbiome data much less useful because these fairly large animals were likely stressed by transport and being held in extremely small containers (1 gallon), w water change 3 times weekly. Microbiomes are very fast-changing and extremely sensitive to environment. Just the transition from field to lab would be concerning, but to transport across country and held in 1 gallon containers I would expect changed microbiomes.

A lot of work went into the transcriptomics and microbiome methods and these methods seem adequate. The problem is that the handling of the animals precludes robust interpretation of these results, so this work is not up to the quality I expect for a Proc B paper.

Based of its limitations, this lab study offers no new insights into the causes of SSWD, although it does try to comment on causes of SSWD. Its main strength is the transcriptional results on immune function. From this lab study, it is precarious and misleading to try and make the following conclusions:

“Taken together, these results suggest that the cause of SSWD may not be a single pathogen as these and other results suggest [17,29], but rather maybe due to a positive feedback loop between abiotic and biotic drivers, where a disturbance of the microbial community, low oxygen conditions, and proliferation of copiotrophic microbes creates an anoxic microenvironment for sea stars leading to hypoxia and loss of collagen control leading to tissue degradation which may further spillover, creating high organic matter, low oxygen conditions for nearby stars [50].

These conclusions would suggest that only stars in hypoxic conditions died from SSWD, and not what occurred: a coast-wide epidemic including extremely pristine, high wave energy environments.

Decision letter (RSPB-2021-0772.R0)

12-May-2021

Dear Dr Pespeni:

I am writing to inform you that we have now obtained responses from referees on manuscript RSPB-2021-0772 entitled "Sea stars resist wasting through an active immune system and a healthy microbiome" which you submitted to Proceedings B.

Unfortunately, on the advice of the Associate Editor and the referees, your manuscript has been rejected following full peer review. Both referees find major problems with interpretation of the microbiome data, and recommend that you focus solely on the transcriptome study. However, such a change would constitute a very different manuscript rather than a 'major revision'. Competition for space in Proceedings B is currently extremely severe, as many more manuscripts are submitted to us than we have space to print. We are therefore only able to publish those that are exceptional, convincing and present significant advances of broad interest, and must reject many good manuscripts.

Please find below the comments received from referees concerning your manuscript, not including confidential reports to the Editor. I hope you may find these useful should you wish to submit your manuscript elsewhere.

We are sorry that your manuscript has had an unfavourable outcome, but would like to thank you for offering your work to Proceedings B.

Best wishes,
Innes Cuthill

Professor Innes Cuthill
mailto:proceedingsb@royalsociety.org

Associate Editor

Board Member: 1

Comments to Author:

After reviewing the paper and both reviews, I do not think this paper can be published in the RCN special issue. Both reviewers identified serious issues with both the microbiome (reviewer 1) and transcriptomic components (reviewer 2) of the study that disqualify it from publication at this time. As noted by reviewers, after addressing these issues, it is possible this study may be published in a more specialized marine journal. I am sorry that I do not have better news for the authors.

Reviewer(s)' Comments to Author:

Referee: 1

Comments to the Author(s)

General Comments

Sea star wasting disease has gained prominence in recent years due to its widespread mortality on the North American Pacific Coast. The etiology of the condition is not fully resolved, and many environmental or microbial insults result in similar disease signs, but no single factor can consistently explain wasting outbreaks. Lloyd and Pespeni extend on a previous study (Lloyd & Pespeni, 2018 Sci. Rep), which surveyed microbiome composition during wasting progression, to look at host gene expression, focusing especially on genes involved in collagen and hypoxia response which have recently been suggested as disease triggers. They found that the expression of genes involved in collagen (presumably synthesis) was greater in specimens that did not waste during their 18 d experiment than in specimens that wasted, and found evidence that hypoxia-response (notably HIF1a) had greater expression in animals that did waste. They extend transcriptome information to previously sequenced 16S rRNAs to identify linkage between these components and conclude that greater abundance of 'beneficial' microorganisms in healthy specimens transitioned to more 'pathogenic' and 'copiotrophic' taxa as individuals wasted.

There is really useful and interesting data on host gene transcription buried in this manuscript – the author’s finding of collagen- and hypoxia-response gene patterns echo recent work showing that low oxygen conditions generated at the animal’s surface may relate to wasting response (Aquino et al. 2021), and this falls in line with work showing that wasting is a basal-to-surface process affecting collagen and body wall tissues (see: DOI: <https://doi.org/10.3354/dao03598>). However, this reviewer has significant concerns about the interpretation of the microbiome data presented in this manuscript (and originally presented in the Lloyd and Pespeni 2018 Sci Rep paper). The biggest concerns regarding the microbiome part of this manuscript are:

A. The authors retrieved sea stars from Monterey, California, shipped them to Vermont, and then maintained them in artificial seawater (ASW; Instant Ocean). While ASW can be used to mimic the inorganic constituents of seawater quite well – and metazoans can be maintained in captivity using this medium – it does not mimic naturally occurring organic compounds in seawater.

Marine bacteria are highly sensitive to containment (bottle effects); the surface microbiome of captive sea stars likely mostly represents taxa that consume mucus and other substrates put out by the sea stars themselves – rather than those that are firmly embedded in tissues. There is every expectation that these will change rapidly if sea stars are transferred into ASW (see Kooperman et al., 2007 FEMS Microb Lett; Friel et al., 2020 Front Micro). This reviewer disagrees that host-associated microbiomes, especially surface-associated microorganisms can be studied using samples maintained in ASW for this reason. Host gene transcription, on the other hand, is likely less affected by ASW conditions.

B. The authors analyzed microbiome data by performing RT-PCR on RNA extracts, rather than PCR of DNA extracts. These answer two different questions. rRNAs (c.f. rDNA) levels vary between taxa primarily by their relative activity but also growth strategies (see Steven et al., 2017 Applied and Environmental Microbiology; Bowsher et al., 2019 mSystems). It is possible, for example, for numerically rare taxa to be fast growing and produce lots of rRNA, but numerically abundant taxa to be slower-growing and produce fewer rRNAs (see Gremoin et al., 2003 Environmental Microbiology). In a community sense, rRNA tells you the relative activity of different constituents and rDNA tells you (with PCR bias caveats) numerical relative abundance. For this reason, rRNA-based approaches cannot provide relative abundance data for microbial communities.

C. The authors never qualify satisfactorily trait-based assignments to 16S rRNA matches. Just because a bacterial genus has pathogenic or “beneficial” representative does not mean that it is possible to say that these genera all share that trait – in fact, for most bacterial genera, pathogens are the exception to the rule for genus-level assignments. I’m still not sure how “known beneficial” could be derived from previous work, unless there were co-culture based studies which illustrated enhanced metazoan growth rates or immune function, etc. An alternate approach to labeling bacterial taxa with these traits is to focus on their relative growth capacity based on genome comparisons – see for example Haggerty & Dinsdale, 2016 Global Ecol & Biogeog. Unless taxonomic assignments are made at 100% similarity across the 16S rRNA, this reviewer does not believe assignment to “pathogen” is possible – and even then, microorganisms that are 100% the same across this gene can have vastly different gene inventories (see Jaspers & Overmann, 2004, Appl Env Micro).

On the other hand, the data on host gene transcription is interesting and, in this reviewer’s opinion, meriting publication on its own, either by removing the microbiome data altogether, or reducing it and focusing on general growth strategies of bacterial genera – and acknowledging limitation of aquarium-based studies and their impacts on microbiome constituents. I’ve made many suggestions below which I hope assist the authors in a revision.

Line 11: I recommend replacing “healthy” and “sick” with their veterinary terms “grossly normal” (or “not wasting”) and “wasting-affected” here and throughout the manuscript

Line 12: Were the “collagen” genes actual genes involved in collagen generation, or were these collagenases or remodeling genes? It would probably be best to go with a gene functional ontology which is consistent with other studies.

Line 13: Same comment here: “hypoxia-related” genes – are these response to hypoxia or caused by hypoxia or...? A little later in the manuscript I read that these include “hypoxia inducible factor” – if this is the only hypoxia-related gene I would recommend keeping it as “hypoxia inducible factor”

Line 16: The authors list here and throughout the manuscript three “known” traits for genera, including “beneficial”, “pathogenic” and “copiotrophic”. It’s unclear how or who established these traits for all species within these genera – i.e. how are these “known”?

Line 17-18: “... considering genotypes of stars at 98,145 SNPs, no variants...” this doesn’t make sense. Perhaps replace “at” with “had” as appropriate

Lines 24 – 35: While I think this is well put and a nice summary of infectious diseases, it sets up the manuscript as an investigation of an infectious disease. There is currently no evidence that this is the case for sea star wasting disease, or any echinoderm-related disease as well. Rather, these may be environmentally-cued autocatalytic stress responses. See for example Delroisse et al., 2020 Sci Rep and sea cucumber skin ulceration disease.

Lines 67 – 69: Again, this statement implicates that there is a pathogen involved in sea star wasting – it might be better to state this as “could provide insight into the role of microorganisms in, and reveal host responses to, disease”

Line 71 – 73: See above comment about the limitations of aquarium-based studies. I would also caution against the identification of “healthy” microbiomes – rather, these are microbiomes that are observed before wasting occurs.

Line 87 – 88: Were the specimens shipped in water? On ice?

Lines 93 – 94: Instant ocean is a mix of inorganic salts and other ions that mimics the chemical environment of seawater, but as mentioned above does not mimic organic constituents of seawater on which heterotrophic bacteria rely – for example, coral reef aquariums are typically amended with carbon sources since they are carbon limited. Bacterial/ archaeal communities inhabiting aquariums are very different from natural environments (unless they are symbiotic/embedded within tissues). Most bacteria in biopsy punches are probably at the animal’s surface, which will change greatly in captivity

Line 96: Biopsy punches were taken every 3 days for 15 days over the course of the experiment. How do these line up with the appearance of wasting? For example, if wasting occurred half-way through these 3 days, then which biopsy was used? This raises an important consideration when comparing early vs late wasting – if wasting is observed a day after a punch, and then considered early wasting, and in another star wasting starts on the day that a punch is taken and also considered early wasting, both gene expression and microbiome constituents reflect different stages of disease progression.

Line 121: Please include “16S rRNA amplicon” – not just “16S amplicon”. See above for comment on basing quantitative microbiome interpretations on RNA- (vs. DNA-) template libraries.

Line 127: A conceptual diagram showing when animals became sick and when transcriptomic/16S rRNA amplicon library samples were collected would be helpful. Did all “sick” specimens do so at the same time? Line 128: So all analyses were performed on singular specimens? (“from a healthy individual”).

Line 153 – Line 155: If I understand this analysis correctly, this would seemingly be a quantitative interpretation of bacterial OTUs. Because of the problems in interpretation of RNA-based libraries, it would be more appropriate to use non-quantitative statistics on their presence/absence (of ‘active’ bacteria).

Line 160 – 161: “at first onset of SSWD” – how was this captured when biopsy punches were taken every 3 days?

Line 166: “immediately before and after” – so this means that onset occurred within these 3 day windows?

Lines 191 – 204: See above – the authors should explore transcript read count vs OTU presence/absence.

Line 240: 6×10^4 to 4×10^7 is a huge range of variation of reads per sample. Was any rarefaction performed to normalize edge effects?

Lines 251 – 254: The presentation of broad categories (ontologies?) in the text is discordant with Figure 1. Figure 1a in the copy I review shows three immune-related genes; Figure 1b shows three transcription/translated related genes. The text mentions structural related genes, and it's unclear which of the ontologies in this figure correspond with this category. Likewise, the figure identifies collagen-genes, but I can't find what genes these actually are in either the figure or elsewhere in the text.

Line 264-265 and Fig. 2: So the only hypoxia related gene is HIF 1a – so it would be better to state this throughout the manuscript instead of “hypoxia-related genes” (plural). Was collagenase 3 the only collagen-related gene? I think “collagen-degradation and remodeling gene” would be a more accurate moniker.

Line 301: The sentence beginning with “While” should perhaps start with “Seven”.

Line 305: So if the “collagen-related” genes were collagenases, and they were more expressed in healthy c.f. wasting stars, this would mean that collagen is being broken down, potentially, more in healthy stars than wasting stars. If there are more genes within this ontology – for example, collagen genesis, then I'd recommend splitting these out to reflect constructive and destructive processes.

Line 310: “genes related to collagen production” – ah, so this includes both production and destructive processes. I would recommend splitting these. Are these genes all echinoderm transcripts or might these include bacterial transcripts as well? Even with polyA selection, there is always some contamination with prokaryotic transcripts within transcriptomes.

Lines 308 – 316: The comparison between healthy and late-stage disease that shows an upregulation of immune genes is not surprising, since at this stage the specimens are infiltrated by bacteria and other microorganisms through lesions.

What would really help in this list of up- and down-regulated transcripts could be a conceptual figure showing a timeline of disease progression with the various categories.

Line 331: While it is certainly interesting to see trends in microbiome composition and correlations to host transcripts, there could well be correlation between un-related processes. Add to this the difficulty in quantitative interpretation based on RNA libraries, and it is this reviewer's opinion that this part of the manuscript does not add much to knowledge of disease process.

Line 358: The last sentence of this paragraph is not complete

Line 368 – 369: Sea star wasting disease has never been shown to be highly transmissible.

Line 372 – 375: Decreases in relative abundance during disease progression could relate to increases in absolute abundance of other groups – i.e. they may just grow more slowly. It is not

appropriate in this reviewer's mind to attribute these as "healthy" or "beneficial" microbiome constituents without rigorous testing to compare health states with and without these organisms. At the very least, quantitative (vs relative i.e. amplicon libraries) approaches are necessary to examine their dynamics during disease progression before these can be identified with any trait.

Line 385 – 387: It is not exactly a hypothesis that there are some microbiome constituents correlate with patterns of gene expression in the host.

Lines 400 – 410: Much of this is a restatement of the results. It would be more appropriate to focus on the authors interpretation (i.e. lines 408 – 410), and place these at the start.

Lines 434 – 450: Most of this section is speculation. Sure, there are species within the genera identified that can produce e.g. antibiotic and antimicrobial compounds, or may be pathogenic, but extending this to the broad number of taxa within a single genus is highly tenuous.

Line 454: Unfortunately, the evidence for a healthy microbiome implicated in constantly healthy specimens is not warranted by the data provided (alone).

Referee: 2

Comments to the Author(s)

Thirty eight asymptomatic *Pisaster ochraceus* seastars were flown from Monterey California to Burlington Vt and held in 1 gallon containers w artificial seawater (3 water changes/ week). IN a 15 day lab observation, 16 stars became sick and 8 remained healthy. What happened with the other 14 that were shipped? Were they dead or sick on arrival, suggesting the transit was stressful? IN the Results, line 1 it says: During the two-week experiment, 29 individuals showed signs of SSWD and 8 remained healthy.

The surface microbiome and immune related genes were monitored in 24 of the stars for 15 days. Why was the experiment terminated at 15 days? Because more and more stars were becoming sick and dying? Inferences were made about causative agent and immune function based on comparing whether the outcome was to become sick or remain healthy. Caution is needed in interpreting microbiome trajectories since this is a lab experiment with long-distance shipped animals that might already be stressed by transport and lab conditions. Further caution is needed in inferring cause and effect between immune condition and bacterial surface changes. While the authors conclude that a healthy immune response drives a beneficial microbiome, its also possible that a healthy microbiome is permissive of a healthy immune response.

The study concludes that "Animals that remained healthy had an active immune response and a beneficial microbial community while animals that became sick showed evidence of responding to hypoxia and a proliferation of opportunistic microbes that thrive in carbon-rich, oxygen-poor environments."

The immune data are the most useful. The authors found higher expression of immune-related, tissue integrity and collagen genes in healthy relative to sick individuals. I think the result of lower expression of collagen-related genes in sick relative to healthy is interesting and deserves further investigation. The finding of higher expression of hypoxia-related genes in sick relative to healthy individuals is harder to interpret, since this could be an artifact of lab stress conditions.

I find the microbiome data much less useful because these fairly large animals were likely stressed by transport and being held in extremely small containers (1 gallon), w water change 3 times weekly. Microbiomes are very fast-changing and extremely sensitive to environment. Just the transition from field to lab would be concerning, but to transport across country and held in 1 gallon containers I would expect changed microbiomes.

A lot of work went into the transcriptomics and microbiome methods and these methods seem adequate. The problem is that the handling of the animals precludes robust interpretation of these results, so this work is not up to the quality I expect for a Proc B paper.

Based on its limitations, this lab study offers no new insights into the causes of SSWD, although it does try to comment on causes of SSWD. Its main strength is the transcriptional results on immune function. From this lab study, it is precarious and misleading to try and make the following conclusions:

“Taken together, these results suggest that the cause of SSWD may not be a single pathogen as these and other results suggest [17,29], but rather maybe due to a positive feedback loop between abiotic and biotic drivers, where a disturbance of the microbial community, low oxygen conditions, and proliferation of copiotrophic microbes creates an anoxic microenvironment for sea stars leading to hypoxia and loss of collagen control leading to tissue degradation which may further spillover, creating high organic matter, low oxygen conditions for nearby stars [50].

These conclusions would suggest that only stars in hypoxic conditions died from SSWD, and not what occurred: a coast-wide epidemic including extremely pristine, high wave energy environments.

Author's Response to Decision Letter for (RSPB-2021-0772.R0)

See Appendix A.

RSPB-2022-2277.R0

Review form: Reviewer 2

Recommendation

Major revision is needed (please make suggestions in comments)

Scientific importance: Is the manuscript an original and important contribution to its field?

Marginal

General interest: Is the paper of sufficient general interest?

Marginal

Quality of the paper: Is the overall quality of the paper suitable?

Marginal

Is the length of the paper justified?

Yes

Should the paper be seen by a specialist statistical reviewer?

Yes

Do you have any concerns about statistical analyses in this paper? If so, please specify them explicitly in your report.

No

It is a condition of publication that authors make their supporting data, code and materials available - either as supplementary material or hosted in an external repository. Please rate, if applicable, the supporting data on the following criteria.

Is it accessible?

N/A

Is it clear?

N/A

Is it adequate?

N/A

Do you have any ethical concerns with this paper?

No

Comments to the Author

The goal of this study was to compare the gene expression patterns of one species of seastar (*Pisaster ochraceus*) affected by the catastrophic sea star wasting disease epidemic. The authors measured gene expression longitudinally of 24 sea stars (*Pisaster ochraceus*), collected from Monterey and shipped to culture in an artificial seawater system in Vermont. 8 individual stars remained asymptomatic and sixteen progressed through stages of SSW over 2 weeks. The rapidity of disease onset in two thirds of the test subjects suggests they were stressed by some combination of shipping and subsequent culture conditions.

Pespeni and Lloyd conclude from longitudinal gene expression patterns that asymptomatic animals remained healthy using an active immune response and that both early- and late-stage wasting animals lost the regulation of systems that maintain tissue integrity.

Their assembled transcriptome, genes of interest, and sequence data files make for useful resources for future work related to Sea Star Wasting. Particularly interesting is the result that previously identified immune, tissue integrity, and pro-collagen genes were more highly expressed in asymptomatic relative to wasting individuals. In wasting stars, hypoxia-inducible factor 1-alpha and RNA processing genes were more highly expressed. It is noteworthy that the expression data indicate compromise of the asteroid catch collagen system, followed by lesions and loss of tissue integrity.

The transcriptomic work and analyses are done with an adequate sample size. There is high variability in the data as seen in the large data range and error bars in the longitudinal data plots Fig 4C. I can't tell if this level of variability requires any different statistical handling.

There are some weaknesses in the study:

It is likely that epidemics of aquaculture species, like shrimp viruses (and perhaps salmon viruses) would be considered larger panzootics than SSW, so it does not seem correct to say on Line 32 this is the largest marine epidemic... It is likely to be correct to follow others in specifying it is largest panzootic of marine wildlife.

It seems inappropriate and incorrect to refer to non-diseased stars as healthy under these conditions. It is not correct to call the asymptomatic animals healthy, since nothing is actually known about their health state. Asymptomatic as compared to symptomatic would better reflect the differences between stars that developed signs of disease and those that did not. What was the final outcome for these animals after the 2 week study period? Did the 8 "asymptomatic" stars become sick or die? This gets back to concern that its not correct to call these animals "healthy", all that can be observed is that they are asymptomatic. I regret if I missed seeing this information, but did you report the level of mortality or arm loss in the different treatments?

One gallon containers with no active water flow seem very small for *Pisaster ochraceus* unless these were v small individuals? What were average diameters of the test animals? I am amazed that so many survived in small containers with water changed only every 3 days... The combination of stressful shipping and this kind of lab culture does concern me about the greater relevance of the gene expression results. It seems likely that these kind of stressful conditions explain why two thirds of the test animals developed wasting.

It's important that for this paper it is made clear that these results are very specific to their scenario and conditions and these results may not be applicable to what's happening in nature or even in lab-held animals kept under better conditions.

Please specify what species of sea star Gudenkauf and Hewson used in their transcriptomic study.

a few typos/extra words:

- o Line 266: period after "healthy than." is not needed
- o Lines 277 and 292: "collengenase" should be "collagenase"?
- o Line 333: don't need the word "in" before the word "(orange)"

Review form: Reviewer 3

Recommendation

Accept as is

Scientific importance: Is the manuscript an original and important contribution to its field?

Excellent

General interest: Is the paper of sufficient general interest?

Excellent

Quality of the paper: Is the overall quality of the paper suitable?

Excellent

Is the length of the paper justified?

Yes

Should the paper be seen by a specialist statistical reviewer?

No

Do you have any concerns about statistical analyses in this paper? If so, please specify them explicitly in your report.

No

It is a condition of publication that authors make their supporting data, code and materials available - either as supplementary material or hosted in an external repository. Please rate, if applicable, the supporting data on the following criteria.

Is it accessible?

N/A

Is it clear?

N/A

Is it adequate?

N/A

Do you have any ethical concerns with this paper?

No

Comments to the Author

This paper provides critical information to help us understand a dramatic and barely understood marine epidemic, the largest marine wildlife epidemic ever documented. The organisms involved, sea stars, have historically received remarkably little research attention, despite their critical roles in ecosystems, their phylogenetic affinity as a deuterostomes, and their enduring popular appeal. Experiments with sea stars and wasting disease are exceedingly few and the microbiome is an excellent if challenging place to look for disease relationships.

The paper is greatly improved with the critical eyes of reviewers' and editor's many suggestions that have clarified and highlighted what is known and not known. It is of further great interest how the authors carried out the lab microbiome experiments without seeing substantive control treatment changes as well...that is also well worth reporting as they have done in the revision. A little further info here would be helpful. Specifically, where in particular were the stars collected in Monterey? This would be helpful background info for future comparisons and for knowledge about how that particular population has been faring with SSWD, in general. Also, I assume the stars were not fed during the experiment, but perhaps that should be stated.

Decision letter (RSPB-2022-2277.R0)

03-Feb-2023

I am writing to inform you that this version of your manuscript RSPB-2022-2277 entitled "Sea stars resist wasting through active immune and collagen systems" has, in its current form, been rejected for publication in Proceedings B.

This action has been taken on the advice of referees, who have recommended that substantial revisions are necessary. With this in mind we would be happy to consider a resubmission, provided the comments of the referees are fully addressed. However please note that this is not a provisional acceptance.

- 1) A 'response to referees' document including details of how you have responded to the comments, and the adjustments you have made.
- 2) A clean copy of the manuscript and one with 'tracked changes' indicating how you addressed the editors and referees' comments.
- 3) Line numbers in your main document.
- 4) Please read our data sharing policies to ensure that you meet our requirements <https://royalsocietypublishing.org/journals/authors/author-guidelines/#data>.

Sincerely,
Dr Sasha Dall
mailto:proceedingsb@royalsociety.org

Reviewer(s)' Comments to Author:
Referee: 2
Comments to the Author(s).

The goal of this study was to compare the gene expression patterns of one species of seastar (*Pisaster ochraceus*) affected by the catastrophic sea star wasting disease epidemic. The authors measured gene expression longitudinally of 24 sea stars (*Pisaster ochraceus*), collected from Monterey and shipped to culture in an artificial seawater system in Vermont. 8 individual stars remained asymptomatic and sixteen progressed through stages of SSW over 2 weeks. The rapidity of disease onset in two thirds of the test subjects suggests they were stressed by some combination of shipping and subsequent culture conditions.

Pespeni and Lloyd conclude from longitudinal gene expression patterns that asymptomatic animals remained healthy using an active immune response and that both early- and late-stage wasting animals lost the regulation of systems that maintain tissue integrity.

Their assembled transcriptome, genes of interest, and sequence data files make for useful resources for future work related to Sea Star Wasting. Particularly interesting is the result that previously identified immune, tissue integrity, and pro-collagen genes were more highly expressed in asymptomatic relative to wasting individuals. In wasting stars, hypoxia-inducible factor 1-alpha and RNA processing genes were more highly expressed. It is noteworthy that the expression data indicate compromise of the asteroid catch collagen system, followed by lesions and loss of tissue integrity.

The transcriptomic work and analyses are done with an adequate sample size. There is high variability in the data as seen in the large data range and error bars in the longitudinal data plots Fig 4C. I can't tell if this level of variability requires any different statistical handling.

There are some weaknesses in the study:

It is likely that epidemics of aquaculture species, like shrimp viruses (and perhaps salmon viruses) would be considered larger panzootics than SSW, so it does not seem correct to say on Line 32 this is the largest marine epidemic... It is likely to be correct to follow others in specifying it is largest panzootic of marine wildlife.

It seems inappropriate and incorrect to refer to non-diseased stars as healthy under these conditions. It is not correct to call the asymptomatic animals healthy, since nothing is actually known about their health state. Asymptomatic as compared to symptomatic would better reflect the differences between stars that developed signs of disease and those that did not. What was the final outcome for these animals after the 2 week study period? Did the 8 "asymptomatic" stars become sick or die? This gets back to concern that its not correct to call these animals "healthy", all that can be observed is that they are asymptomatic. I regret if I missed seeing this information, but did you report the level of mortality or arm loss in the different treatments? One gallon containers with no active water flow seem very small for *Pisaster ochraceus* unless these were v small individuals? What were average diameters of the test animals? I am amazed that so many survived in small containers with water changed only every 3 days... The combination of stressful shipping and this kind of lab culture does concern me about the greater

relevance of the gene expression results. It seems likely that these kind of stressful conditions explain why two thirds of the test animals developed wasting.

It's important that for this paper it is made clear that these results are very specific to their scenario and conditions and these results may not be applicable to what's happening in nature or even in lab-held animals kept under better conditions.

Please specify what species of sea star Gudenkauf and Hewson used in their transcriptomic study.

a few typos/extra words:

- o Line 266: period after "healthy than." is not needed
- o Lines 277 and 292: "collengenase" should be "collagenase"?
- o Line 333: don't need the word "in" before the word "(orange)"

Referee: 3

Comments to the Author(s).

This paper provides critical information to help us understand a dramatic and barely understood marine epidemic, the largest marine wildlife epidemic ever documented. The organisms involved, sea stars, have historically received remarkably little research attention, despite their critical roles in ecosystems, their phylogenetic affinity as a deuterostomes, and their enduring popular appeal. Experiments with sea stars and wasting disease are exceedingly few and the microbiome is an excellent if challenging place to look for disease relationships.

The paper is greatly improved with the critical eyes of reviewers' and editor's many suggestions that have clarified and highlighted what is known and not known. It is of further great interest how the authors carried out the lab microbiome experiments without seeing substantive control treatment changes as well...that is also well worth reporting as they have done in the revision. A little further info here would be helpful. Specifically, where in particular were the stars collected in Monterey? This would be helpful background info for future comparisons and for knowledge about how that particular population has been faring with SSWD, in general. Also, I assume the stars were not fed during the experiment, but perhaps that should be stated.

Author's Response to Decision Letter for (RSPB-2022-2277.R0)

See Appendix B.

RSPB-2023-0347.R0

Review form: Reviewer 4

Recommendation

Accept as is

Scientific importance: Is the manuscript an original and important contribution to its field?

Good

General interest: Is the paper of sufficient general interest?

Good

Quality of the paper: Is the overall quality of the paper suitable?

Good

Is the length of the paper justified?

Yes

Should the paper be seen by a specialist statistical reviewer?

No

Do you have any concerns about statistical analyses in this paper? If so, please specify them explicitly in your report.

No

It is a condition of publication that authors make their supporting data, code and materials available - either as supplementary material or hosted in an external repository. Please rate, if applicable, the supporting data on the following criteria.

Is it accessible?

Yes

Is it clear?

Yes

Is it adequate?

Yes

Do you have any ethical concerns with this paper?

No

Comments to the Author

This manuscript explores the gene expression of *Pisaster ochraceus* in the lab, comparing individuals that spontaneously wasted following collection and transportation to lab conditions to those that remained asymptomatic across the same conditions. Within the spontaneously wasting group, gene expression was compared between different stages of wasting. In addition, the manuscript compares gene expression and microbiome abundance correlations within the same dataset. The authors found that there was differential gene expression throughout the compared stages of wasting, with asymptomatic individuals showing increased expression of genes related to tissue integrity, collagen and immune responses, and symptomatic individuals showing increased gene expression of RNA processing genes and hypoxia related genes. They found that microbiomes associated to asymptomatic individuals remained relatively stable (with some changes over time), and that there were some microbial species that were associated with symptomatic individuals. The authors additionally found no genotype variants that were associated with symptomatic status in the experiment.

I think this was a clearly written manuscript, with elegant figures that provides important information on a relevant and interesting question. There were a few minor points that I think could be addressed, including a few typos, which are outlined below.

Likely while this manuscript has been in review, and additional manuscript was published that looks at wild *P. ochraceus* microbiomes and it would be worth incorporating some short points from Loudon et al, 2023 into the discussion.

Loudon, A. H., J. Park, and L. W. Parfrey. 2023. Identifying the core microbiome of the sea star *Pisaster ochraceus* in the context of sea star wasting disease. *FEMS Microbiology Ecology*.

Line edits:

Line 98: This is a minor point, but stating that Monterey 'in the center' of the species range seems like a pretty broad statement. Since the range is Baja to AK, it would depend on how wide a range you consider 'the center' whether that statement is true. Can you either soften the language or cite the sources for Monterey being considered the center?

Line 188: I think there is a word missing here

Figure 1D: why did you choose to combine all samples here, and not split between asymptomatic and wasting, as is done in the other panels? Especially since the statistic is about the comparison between wasting and asymptomatic, not between collagen and non-collagen expression.

Line 311-312: I am not sure why 'in the common environmental conditions of individual lab aquaria' is included in this statement of the result. It seems odd to point that out here, when it is likely true of all other results as well.

Line 464-465: 'cell adhesion' is currently referenced twice.

Line 484: Missing a word

Line 517-518: How sure can you be that all stars in your experiments were exposed to wasting? Were they ever co-housed while some stars showed signs of wasting? Are you assuming they were all exposed because they were collected from the same location? This is a different framing then has been used through the manuscript and makes a claim that is a bit beyond what you currently have support for.

Review form: Reviewer 3

Recommendation

Accept as is

Scientific importance: Is the manuscript an original and important contribution to its field?

Excellent

General interest: Is the paper of sufficient general interest?

Excellent

Quality of the paper: Is the overall quality of the paper suitable?

Excellent

Is the length of the paper justified?

Yes

Should the paper be seen by a specialist statistical reviewer?

No

Do you have any concerns about statistical analyses in this paper? If so, please specify them explicitly in your report.

No

It is a condition of publication that authors make their supporting data, code and materials available - either as supplementary material or hosted in an external repository. Please rate, if applicable, the supporting data on the following criteria.

Is it accessible?

Yes

Is it clear?

Yes

Is it adequate?

Yes

Do you have any ethical concerns with this paper?

No

Comments to the Author

Changes have improved the ms.

A few further small notes for clarification in the methods:

Pg 3, line 40: not reduction in habitat, but rather, alteration in habitat

Pg 5, line 89: "the wharf at Monterey Harbor". Please be more specific, by telling us which wharf, the depth range, and something about the habitat

Pg 6-7, what criteria were used to select the particular 24 out of 38 stars?
What was the size range of the stars?

How deep is the biopsy punch? Does it go all the way through the body wall and into the coelomic cavity? Punch into organs?

Decision letter (RSPB-2023-0347.R0)

11-Apr-2023

Dear Dr Pespeni

I am pleased to inform you that your manuscript RSPB-2023-0347 entitled "Sea stars resist wasting through active immune and collagen systems" has been accepted for publication in Proceedings B.

The referee(s) have recommended publication, but also suggest some minor revisions to your manuscript. Therefore, I invite you to respond to the referee(s)' comments and revise your manuscript. Because the schedule for publication is very tight, it is a condition of publication that you submit the revised version of your manuscript within 7 days. If you do not think you will be able to meet this date please let us know.

When submitting your revision please upload a file under " to Referees" - in the "File Upload" section. This should document, point by point, how you have responded to the reviewers' and Editors' comments, and the adjustments you have made to the manuscript. We also require a copy of the revised manuscript showing track changes to be uploaded.

4) Data accessibility section and data citation

It is a condition of publication that data supporting your paper are made available either in the electronic supplementary material. Authors must complete the 'data accessibility' section in the submission system. This should list the database and accession number for all data from the article that has been made publicly available, for instance:

NB. From April 1 2013, peer reviewed articles based on research funded wholly or partly by RCUK must include, if applicable, a statement on how the underlying research materials – such as data, samples or models – can be accessed.

[http://datadryad.org/submit?journalID=RSPB&manu=\(Document not available\)](http://datadryad.org/submit?journalID=RSPB&manu=(Document not available)) which will take you to your unique entry in the Dryad repository. If you have already submitted your data to dryad you can make any necessary revisions to your dataset by following the above link.

Please include the Dryad DOI in the Data Accessibility section and reference in the paper's bibliography.

Please see our Data Sharing Policies (<https://royalsociety.org/journals/authors/author-guidelines/>).

6) A media summary: a short non-technical summary (up to 100 words) of the key findings/importance of your manuscript.

Sincerely,
Dr Sasha Dall
mailto:proceedingsb@royalsociety.org

Reviewer(s)' Comments to Author:

Referee: 4

Comments to the Author(s).

This manuscript explores the gene expression of *Pisaster ochraceus* in the lab, comparing individuals that spontaneously wasted following collection and transportation to lab conditions to those that remained asymptomatic across the same conditions. Within the spontaneously wasting group, gene expression was compared between different stages of wasting. In addition, the manuscript compares gene expression and microbiome abundance correlations within the same dataset. The authors found that there was differential gene expression throughout the compared stages of wasting, with asymptomatic individuals showing increased expression of genes related to tissue integrity, collagen and immune responses, and symptomatic individuals showing increased gene expression of RNA processing genes and hypoxia related genes. They found that microbiomes associated to asymptomatic individuals remained relatively stable (with some changes over time), and that there were some microbial species that were associated with symptomatic individuals. The authors additionally found no genotype variants that were associated with symptomatic status in the experiment.

I think this was a clearly written manuscript, with elegant figures that provides important information on a relevant and interesting question. There were a few minor points that I think could be addressed, including a few typos, which are outlined below.

Likely while this manuscript has been in review, and additional manuscript was published that looks at wild *P. ochraceus* microbiomes and it would be worth incorporating some short points from Loudon et al, 2023 into the discussion.

Loudon, A. H., J. Park, and L. W. Parfrey. 2023. Identifying the core microbiome of the sea star *Pisaster ochraceus* in the context of sea star wasting disease. *FEMS Microbiology Ecology*.

Line edits:

Line 98: This is a minor point, but stating that Monterey 'in the center' of the species range seems like a pretty broad statement. Since the range is Baja to AK, it would depend on how wide a range you consider 'the center' whether that statement is true. Can you either soften the language or cite the sources for Monterey being considered the center?

Line 188: I think there is a word missing here

Figure 1D: why did you choose to combine all samples here, and not split between asymptomatic and wasting, as is done in the other panels? Especially since the statistic is about the comparison between wasting and asymptomatic, not between collagen and non-collagen expression.

Line 311-312: I am not sure why 'in the common environmental conditions of individual lab aquaria' is included in this statement of the result. It seems odd to point that out here, when it is likely true of all other results as well.

Line 464-465: 'cell adhesion' is currently referenced twice.

Line 484: Missing a word

Line 517-518: How sure can you be that all stars in your experiments were exposed to wasting? Were they ever co-housed while some stars showed signs of wasting? Are you assuming they were all exposed because they were collected from the same location? This is a different framing then has been used through the manuscript and makes a claim that is a bit beyond what you currently have support for.

Referee: 3

Comments to the Author(s).

Changes have improved the ms.

A few further small notes for clarification in the methods:

Pg 3, line 40: not reduction in habitat, but rather, alteration in habitat

Pg 5, line 89: "the wharf at Monterey Harbor". Please be more specific, by telling us which wharf, the depth range, and something about the habitat

Pg 6-7, what criteria were used to select the particular 24 out of 38 stars?
What was the size range of the stars?

How deep is the biopsy punch? Does it go all the way through the body wall and into the coelomic cavity? Punch into organs?

Author's Response to Decision Letter for (RSPB-2023-0347.R0)

See Appendix C.

Decision letter (RSPB-2023-0347.R1)

01-Jun-2023

Dear Dr Pespeni

I am pleased to inform you that your manuscript entitled "Sea stars resist wasting through active immune and collagen systems" has been accepted for publication in Proceedings B.

You can expect to receive a proof of your article from our Production office in due course, please check your spam filter if you do not receive it. PLEASE NOTE: you will be given the exact page length of your paper which may be different from the estimation from Editorial and you may be asked to reduce your paper if it goes over the 12 page limit.

If you are likely to be away from e-mail contact during this period, let us know. Due to rapid publication and an extremely tight schedule, if comments are not received, we may publish the paper as it stands.

Your article has been estimated as being 12 pages long. Our Production Office will be able to confirm the exact length at proof stage.

Data Accessibility section

Open access

The open access fee is £1,700 per article (plus VAT for UK residents). Payment of open access fees will enable your article to be made freely available via the Royal Society website as soon as it is ready for publication. For more information about open access publishing please visit our website at <https://royalsociety.org/journals/open-access/>. If you have opted for Open Access in Proceedings B, payment of an article processing charge (APC) may be due before your article is published. Our partner Copyright Clearance Center's RightsLink for Scientific Communications will contact the contact or nominated author about your open access options from the email domain @copyright.com (if you have any queries regarding fees, please see <https://royalsocietypublishing.org/rspb/for-authors#question12> or contact authorfees@royalsociety.org). If you now wish to opt for open access then please let us know as soon as possible.

Please note if you are the corresponding author from one of our Royal Society transformative agreements your fees may be covered.

Page charges (for non-Open Access papers)

Our partner Copyright Clearance Center's RightsLink for Scientific Communications will contact the contact or nominated author about payment for page charges.

Sincerely,
Proceedings B
<mailto:proceedingsb@royalsociety.org>

Appendix A

14-Jun-2021

Dear Dr Pespeni,

Your appeal has now been considered by the Editor. I am pleased to let you know that on this occasion, the Editor has decided to allow your appeal, and invites you to resubmit your manuscript to the journal. Specific comments from the Associate Editor are included below:

"I admit I am skeptical that they will be able to address the full scope of issues presented by reviewers, but they do indicate that they have at least some data/solutions to address reviewer issues. For example, they suggest they have data from day 0 and day 3 indicating no change in microbiome in artificial seawater -- but is 3 days sufficient time? Afterall, several studies indicate that captivity in artificial seawater DOES affect the microbiome. I also noticed they did not address the issue of rRNA interpretation/inference raised by reviewer 1. "

We appreciate the opportunity to revise and resubmit our manuscript. We believe we have been able to fully address reviewer concerns and suggestions. Indeed, for example, we have included new analyses showing that the laboratory conditions only minimally affect the microbiome of the sea stars, demonstrating resilience and stability of composition from Day 0 to Day 15 (Fig. 5). Regarding sequencing from rRNA as opposed to rDNA, we appreciate the opportunity to address this concern. We specifically chose to sequence from ribosomal RNA instead of ribosomal DNA because we wanted to capture and focus on the bacterial taxa that were alive and growing, particularly in the context of repeated sampling of individuals through time and disease progression. We have clarified this in the text and revised the language throughout the manuscript.

Reviewer(s)' Comments to Author:

Referee: 1

Comments to the Author(s)

General Comments

Sea star wasting disease has gained prominence in recent years due to its widespread mortality on the North American Pacific Coast. The etiology of the condition is not fully resolved, and many environmental or microbial insults result in similar disease signs, but no single factor can consistently explain wasting outbreaks. Lloyd and Pespeni extend on a previous study (Lloyd & Pespeni, 2018 Sci. Rep), which surveyed microbiome composition during wasting progression, to look at host gene expression, focusing especially on genes involved in collagen and hypoxia response which have recently been suggested as disease triggers. They found that the expression of genes involved in collagen (presumably synthesis) was greater in specimens that did not waste during their 18 d experiment than in specimens that wasted, and found evidence that hypoxia-response (notably HIF1a) had greater expression in animals that did waste. They

extend transcriptome information to previously sequenced 16S rRNAs to identify linkage between these components and conclude that greater abundance of 'beneficial' microorganisms in healthy specimens transitioned to more 'pathogenic' and 'copiotrophic' taxa as individuals wasted.

We greatly appreciate the thorough and thoughtful review by this reviewer. We provide point-by-point responses below.

There is really useful and interesting data on host gene transcription buried in this manuscript – the author's finding of collagen- and hypoxia-response gene patterns echo recent work showing that low oxygen conditions generated at the animal's surface may relate to wasting response (Aquino et al. 2021), and this falls in line with work showing that wasting is a basal-to-surface process affecting collagen and body wall tissues (see: DOI: <https://doi.org/10.3354/dao03598>). However, this reviewer has significant concerns about the interpretation of the microbiome data presented in this manuscript (and originally presented in the Lloyd and Pespeni 2018 Sci Rep paper). The biggest concerns regarding the microbiome part of this manuscript are:

A. The authors retrieved sea stars from Monterey, California, shipped them to Vermont, and then maintained them in artificial seawater (ASW; Instant Ocean). While ASW can be used to mimic the inorganic constituents of seawater quite well – and metazoans can be maintained in captivity using this medium – it does not mimic naturally occurring organic compounds in seawater. Marine bacteria are highly sensitive to containment (bottle effects); the surface microbiome of captive sea stars likely mostly represents taxa that consume mucus and other substrates put out by the sea stars themselves – rather than those that are firmly embedded in tissues. There is every expectation that these will change rapidly if sea stars are transferred into ASW (see Kooperman et al., 2007 FEMS Microb Lett; Friel et al., 2020 Front Micro). This reviewer disagrees that host-associated microbiomes, especially surface-associated microorganisms can be studied using samples maintained in ASW for this reason. Host gene transcription, on the other hand, is likely less affected by ASW conditions.

We agree that the potential effect of ASW on microbiome composition is an important concern for this study, particularly given previous results kindly noted by the reviewer that suggest that ASW affects microbiome composition. We have included these citations as motivation for new analyses presented in the manuscript testing for changes in alpha and beta diversity through the full course of the experiment using on the animals that remained healthy. Any changes in microbiome composition in these animals that started and remained asymptomatic would be driven by the change in the environment, being maintained in ASW in individual aquaria. We found minimal changes in alpha and beta diversity through time, except when considering low abundance taxa, which showed a shift by day 6 and stabilized by day 9. We include the new figure as Fig. 5 and text below to the results.

“Stability of the microbiome in artificial seawater

To test if containment in artificial seawater affected the microbiome composition of the sea star epidermis, we tested for changes in microbiome composition through time for the eight sea stars that remained healthy through the duration of the experiment, Day 0

though Day 15. The Day 0 biopsy was on arrival before introduction to artificial sea water and after an overnight trip from Monterey, CA to Burlington, VT and thus would represent the natural microbiome composition. We found that microbial diversity was relatively stable through time with no significant differences in alpha diversity pairwise between days except for between days 0 and 6 and days 6 and 9 (Faith's diversity incorporating phylogenetic distance, pairwise Kruskal-Wallis, $q < 0.05$), suggesting an initial shift in the microbial community between days 0 and 6, but a return to day 0 diversity levels by day 9. Taxonomic composition patterns (beta diversity) mirrored this result (Fig. 5), showing a shift by Day 6 and a return by Day 9 with differences in relative abundance driven by changes in lower abundance taxa. Weighted and unweighted UniFrac tests for differences in community composition (beta diversity) revealed a similar pattern with no differences when considering weighted dissimilarity (PERMANOVA, pseudo-F = 0.938, $p = 0.489$) and differences when considering unweighted dissimilarity (PERMANOVA, pseudo-F = 2.081, $p = 0.001$). Weighted takes into account relative abundance and thus weights differences in more abundant taxa more heavily; in contrast, unweighted treats each taxon the same regardless of abundance and thus rare or lower abundance taxa have a stronger effect on results. Taken together, these results exploring microbial community changes through time for the animals that remained healthy suggest that overall, there was little impact of artificial seawater on microbial community composition with some changes in lower abundance taxa that stabilized midway through the experiment."

Figure 5. Resilience and stability of the microbiome of healthy sea stars through time in laboratory conditions. (A) Time course of microbiome composition of the eight sea stars that remained visibly healthy through the duration of the experiment; (B) minimal changes in community composition in pairwise comparisons across days (Weighted UniFrac PERMANOVA pseudo-F = 0.938, $P = 0.489$).

B. The authors analyzed microbiome data by performing RT-PCR on RNA extracts, rather than PCR of DNA extracts. These answer two different questions. rRNAs (c.f. rDNA) levels vary between taxa primarily by their relative activity but also growth strategies (see Steven et al., 2017 Applied and Environmental Microbiology; Bowsher et al., 2019 mSystems). It is possible, for example, for numerically rare taxa to be fast growing and produce lots of rRNA, but numerically abundant taxa to be slower-growing and produce fewer rRNAs (see Gremoin et al., 2003 Environmental Microbiology). In a community sense, rRNA tells you the relative activity of different constituents and rDNA tells you (with PCR bias caveats) numerical relative abundance. For this reason, rRNA-based approaches cannot provide relative abundance data for microbial communities.

We thank the reviewer for this comment as it gives us the opportunity to clearly motivate our decision to sequence from rRNA instead of rDNA in the manuscript including the citations noted by the reviewer. We agree that within a sample, sequencing from rDNA would more accurately reflect relative abundance of the microbial taxa represented. However, we were specifically interested in changes in growth rates of microbial taxa through time or across disease stages between samples. For this reason, we chose to sequence from rRNA for the very reasons the reviewer describes, to detect microbial taxa whose growth rates were changing through the course of the experiment. Increases or decreases in taxa with changes in growth rates through time or between disease stages were exactly the signatures we sought to detect.

C. The authors never qualify satisfactorily trait-based assignments to 16S rRNA matches. Just because a bacterial genus has pathogenic or “beneficial” representative does not mean that it is possible to say that these genera all share that trait – in fact, for most bacterial genera, pathogens are the exception to the rule for genus-level assignments. I’m still not sure how “known beneficial” could be derived from previous work, unless there were co-culture based studies which illustrated enhanced metazoan growth rates or immune function, etc. An alternate approach to labeling bacterial taxa with these traits is to focus on their relative growth capacity based on genome comparisons – see for example Haggerty & Dinsdale, 2016 Global Ecol & Biogeog. Unless taxonomic assignments are made at 100% similarity across the 16S rRNA, this reviewer does not believe assignment to “pathogen” is possible – and even then, microorganisms that are 100% the same across this gene can have vastly different gene inventories (see Jaspers & Overmann, 2004, Appl Env Micro).

We agree; we have removed all mentions of “pathogenic” or “beneficial” in this manuscript. We do still discuss “copiotrophic” taxa in the discussion only.

On the other hand, the data on host gene transcription is interesting and, in this reviewer’s opinion, meriting publication on its own, either by removing the microbiome data altogether, or reducing it and focusing on general growth strategies of bacterial genera – and acknowledging limitation of aquarium-based studies and their impacts on microbiome constituents. I’ve made many suggestions below which I hope assist the authors in a revision.

We agree that the host gene expression is the most important contribution of this work. For this revision we have decided to keep the integration with the 16S data as we feel it is a unique analysis, using a network clustering approach that 1) yields insight to the sea star wasting disease dynamics and 2) could be a broadly used in other studies that aim to understand the interactions between host and microbes and disease phenotypes. However, that said, we would be willing to excise the 16S data and analyses from the manuscript if the editor and reviewers think that would be best.

Line 11: I recommend replacing “healthy” and “sick” with their veterinary terms “grossly normal” (or “not wasting”) and “wasting-affected” here and throughout the manuscript

We have replaced “sick” with “wasting” throughout and added definitions to the introduction and methods, linking the words “healthy” with “grossly normal” and “not wasting” and “visibly healthy” and “apparently healthy” and connected “wasting” with “wasting-affected.” Given the broad readership of *Proceedings B*, we still use the word “healthy” in the manuscript, however with greater specificity.

Line 12: Were the “collagen” genes actual genes involved in collagen generation, or were these collagenases or remodeling genes? It would probably be best to go with a gene functional ontology which is consistent with other studies.

This is a good question that allows us to expand/clarify. We have added the text below to the methods of the manuscript to address this question for the reviewer and readers:

“We developed a custom list of echinoderm-annotated, collagen-related genes based on blast annotations including the word “collagen,” which included the words “collagen,” “collagenase,” and “collagenous.” We identified 51 transcripts annotated as being related to collagen; all but four of these genes were pro-collagen, i.e., encoding for different collagen isoforms or precursor molecules, while four were annotated as collagenases. We elected to use a custom list based on gene annotations to other echinoderms rather than GO terms here because these annotations would be more accurate to the unique collagen systems of echinoderms relative to GO categories based on dominant model organisms and our interest in the collagen response during SSW.

Line 13: Same comment here: “hypoxia-related” genes – are these response to hypoxia or caused by hypoxia or...? A little later in the manuscript I read that these include “hypoxia inducible factor” – if this is the only hypoxia-related gene I would recommend keeping it as “hypoxia inducible factor”

Similarly, we have added the text to the methods, “Similarly, we identified five genes as including the word “hypoxia,” including the master regulator, Hypoxia-inducible factor 1-alpha (HIF-1 α) and other subunits of the transcriptional complex.” Given the small number of genes related to this hypothesis we revised the language throughout the manuscript to be more specific as suggested and changed to “hypoxia inducible factor” as opposed to “hypoxia-

related.”

Line 16: The authors list here and throughout the manuscript three “known” traits for genera, including “beneficial”, “pathogenic” and “copiotrophic”. It’s unclear how or who established these traits for all species within these genera – i.e. how are these “known”?

We have revised the abstract and removed mention of specific microbial taxa.

Line 17-18: “... considering genotypes of stars at 98,145 SNPs, no variants...” this doesn’t make sense. Perhaps replace “at” with “had” as appropriate

We have revised the sentence to improve clarity, “Lastly, considering genotypes at 98,145 SNPs, we found no variants associated with whether a star remained visibly healthy or wasted.”

Lines 24 – 35: While I think this is well put and a nice summary of infectious diseases, it sets up the manuscript as an investigation of an infectious disease. There is currently no evidence that this is the case for sea star wasting disease, or any echinoderm-related disease as well. Rather, these may be environmentally-cued autocatalytic stress responses. See for example Delroisse et al., 2020 Sci Rep and sea cucumber skin ulceration disease.

Excellent point. We have revised and expanded the first three paragraphs of the introduction to discuss the factors affecting mass mortality events in the oceans, including SSW, and expanded the discussion to include other echinoderm diseases. These changes improve the framing of the context of our study.

Lines 67 – 69: Again, this statement implicates that there is a pathogen involved in sea star wasting – it might be better to state this as “could provide insight into the role of microorganisms in, and reveal host responses to, disease”

We agree and have revised this statement to be more specific as suggested to not implicate that there is a pathogen. “Here we track changes in gene expression through time of 24 *Pisaster ochraceus* sea stars that remained apparently healthy (8 individuals) or progressed through signs of sea star wasting (16 individuals).” We incorporate the suggested text later in the paragraph, “In the present study, we generate host gene expression data and integrate with the previously generated microbiome data from the same tissue samples to provide insight into the role of microorganisms in, and reveal host responses to, disease.”

Line 71 – 73: See above comment about the limitations of aquarium-based studies. I would also caution against the identification of “healthy” microbiomes – rather, these are microbiomes that are observed before wasting occurs.

We agree and we have deleted that clause of the sentence.

Line 87 – 88: Were the specimens shipped in water? On ice?

Thank you for noting the omission; we have now cited the previous study for more details and added important information to the text as follows:

“Thirty eight adult, asymptomatic *P. ochraceus* were collected from Monterey, California (36°36'21.44"N 121°53'23.69"W) on May 4, 2016 or June 8, 2016 and shipped overnight to the University of Vermont with a total travel time of 17 hours as previously described (REF). Briefly, the sea stars were shipped in a styrofoam box with freezer packs and bunched up layers of newspaper to separate the freezer packs from the sea stars. Each star was individually packed in a plastic bag with a wet paper towel and approximately 300 ml of sea water.”

Lines 93 – 94: Instant ocean is a mix of inorganic salts and other ions that mimics the chemical environment of seawater, but as mentioned above does not mimic organic constituents of seawater on which heterotrophic bacteria rely – for example, coral reef aquariums are typically amended with carbon sources since they are carbon limited. Bacterial/archaeal communities inhabiting aquariums are very different from natural environments (unless they are symbiotic/embedded within tissues). Most bacteria in biopsy punches are probably at the animal’s surface, which will change greatly in captivity.

This is an important concern that we have addressed with new analyses, a new objective added to the study, “to test if maintenance in artificial seawater in the lab affected the sea star epidermal microbiome,” and a new figure (Fig. 5), as discussed above in response to the first major comment. In short, in contrast to previous studies, we observe a high degree of stability and resilience in the sea star microbiome despite artificial sea water conditions. We think this is an important new contribution.

Line 96: Biopsy punches were taken every 3 days for 15 days over the course of the experiment. How do these line up with the appearance of wasting? For example, if wasting occurred half-way through these 3 days, then which biopsy was used? This raises an important consideration when comparing early vs late wasting – if wasting is observed a day after a punch, and then considered early wasting, and in another star wasting starts on the day that a punch is taken and also considered early wasting, both gene expression and microbiome constituents reflect different stages of disease progression.

This is a good question, we have clarified and justified our approach in the methods: “Every three days for fifteen days, nonlethal biopsy punches, using the same instrument noted above, were taken from the epidermis of the body wall of each individual. If an individual was displaying wasting, epidermal tissue at the edge of a lesion was sampled. Given the specificity of the biopsies, individuals were classified by the “symptom number” (see below) displayed at the time of sampling, regardless of signs of wasting the day before or after sampling.”

We agree; ideally, higher resolution sampling would be performed, for example, every day, with cameras monitoring each sea star to capture the moment of the appearance of signs of disease. However, this experiment generated almost 100 RNAseq libraries and sequence data, which is a

substantial investment. The tradeoff could have been to sequence fewer individuals or fewer time points, though we view the number of individuals and time course sampling as a unique strength of the study. Lastly, some individual sea stars showed plasticity or recovery in signs of wasting in our experimental conditions (see samples 37 and 38 in the table S1 below). These patterns suggests that signs of wasting can change rapidly and that the signs at the time of biopsy are most relevant to the characterization of a biopsy for the purposes of gene expression and microbiome data analysis.

Table S1. Time course sampling of epidermal biopsies from *Pisaster ochraceus* used for RNAsequencing.

Count	Individual	Trajectory	Day3	Day6	Day9	Day12	Day15	No. samples per individual
1	10	HH	10 5-08 H	10 5-11 H	10 5-14 H	10 5-17 H	10 5-20 H	5
2	24	HH	24 5-08 H	24 5-11 H	24 5-14 H	24 5-17 H	24 5-20 H	5
3	27	HH	27 5-08 H	27 5-11 H	27 5-14 H	27 5-17 H	27 5-20 H	5
4	08	HS	08 5-08 H	08 5-11 S	08 5-14 S	08 5-17 S	08 5-20 S	5
5	09	HS	09 5-08 H		09 5-14 S	09 5-17 S	09 5-20 S	4
6	15	HS	15 5-08 H	15 5-11 H	15 5-14 H	15 5-17 S	15 5-20 S	5
7	19	HS		19 5-11 H	19 5-14 H	19 5-17 H	19 5-20 S	4
8	20	HS	20 5-08 H	20 5-11 H	20 5-14 H	20 5-17 H	20 5-20 S	5
9	03	SS	03 5-08 S	03 5-11 S				2
10	07	SS	07 5-08 S	07 5-11 S				2
11	14	SS	14 5-08 S	14 5-11 S				2
12	22	SS	22 5-08 S	22 5-11 S	22 5-14 S			3
13	23	SS				23 5-17 S	23 5-20 S	2
14	26	SS	26 5-08 S	26 5-11 S				2
15	28	SS	28 5-08 S	28 5-11 S	28 5-14 S	28 5-17 S		4
16	29	SS	29 5-08 S	29 5-11 S	29 5-14 S			3
17	31	HH	31 6-12 H	31 6-15 H	31 6-18 H	31 6-21 H	31 6-24 H	5
18	32	HH	32 6-12 H	32 6-15 H	32 6-18 H	32 6-21 H		4
19	33	HH	33 6-12 H	33 6-15 H	33 6-18 H	33 6-21 H	33 6-24 H	5
20	34	HH	34 6-12 H	34 6-15 H	34 6-18 H	34 6-21 H	34 6-24 H	5
21	35	HH	35 6-12 H	35 6-15 H	35 6-18 H	35 6-21 H		4
22	36	SS	36 6-12 S	36 6-15 S	36 6-18 S			3
23	37	HS	37 6-12 H	37 6-15 S	37 6-18 H	37 6-21 S		4
24	38	HS	38 6-12 H	38 6-15 S	38 6-18 S	38 6-21 H	38 6-24 S	5

Total no. biopsies sequenced: 93

Abbreviations: H - healthy; S - sick (signs of wasting including lesions and loss of turgor).

Line 121: Please include “16S rRNA amplicon” – not just “16S amplicon”. See above for comment on basing quantitative microbiome interpretations on RNA- (vs. DNA-)template libraries.

Corrected as noted.

Line 127: A conceptual diagram showing when animals became sick and when transcriptomic/16S rRNA amplicon library samples were collected would be helpful. Did all “sick” specimens do so at the same time? Line 128: So all analyses were performed on singular specimens? (“from a healthy individual”).

The animals did not all show signs of wasting at the same time. We have created a table showing the samples used for RNAsequencing (Table S1 and see above). 16S rRNA was sequenced from all samples for which RNAsequencing data were generated.

Regarding “a healthy individual,” the text has been clarified: “Through the course of the fifteen-day experiment, 54 of the 93 samples were taken from individuals that appeared grossly normal (referred to as “healthy”) and 39 were taken from individuals presenting with signs of wasting.”

Line 153 – Line 155: If I understand this analysis correctly, this would seemingly be a quantitative interpretation of bacterial OTUs. Because of the problems in interpretation of RNA-based libraries, it would be more appropriate to use non-quantitative statistics on their presence/absence (of ‘active’ bacteria).

Indeed, we used quantitative approaches because we were interested in the changes in growth (hence the use of RNA-based libraries) between samples across disease stages and time. Please see a more thorough response above to comment “B.”

Line 160 – 161: “at first onset of SSWD” – how was this captured when biopsy punches were taken every 3 days?

We clarified to read, “at first experimental detection of signs of wasting.”

Line 166: “immediately before and after” – so this means that onset occurred within these 3 day windows?

Thank you for this question; the writing was not clear. We have clarified as such: “To test for differential expression changes that coincided with the first experimental detection of signs of wasting, we limited the sample set to samples from the first experimental detection of signs of wasting (symptom: 1) and the sampling time point prior (symptom: 0), when no signs of wasting were observed as well as control sample pairs from visibly healthy individuals at the same time points (28 samples: two time points from 14 individual stars, 6 that showed signs of wasting, 8 that did not). ”

Lines 191 – 204: See above – the authors should explore transcript read count vs OTU presence/absence.

This analysis sought to test for associations between changes in microbial growth/abundance with changes in gene expression. This text was added for clarity.

Line 240: 6×10^4 to 4×10^7 is a huge range of variation of reads per sample. Was any rarefaction performed to normalize edge effects?

Yes, samples were normalized for variation in library size/sequencing effort within DESeq2. This detail was added to the methods and the text in the results was clarified.

Lines 251 – 254: The presentation of broad categories (ontologies?) in the text is discordant with Figure 1. Figure 1a in the copy I review shows three immune- related genes; Figure 1b

shows three transcription/translated related genes. The text mentions structural related genes, and it's unclear which of the ontologies in this figure correspond with this category. Likewise, the figure identifies collagen-genes, but I can't find what genes these actually are in either the figure or elsewhere in the text.

The writing here was unclear. The panels Fig 1A and 1B were referring to transcripts differentially expressed, i.e., among the 33 differentially expressed transcripts. The enrichment results were referring to the GO categories. This has been clarified by re-writing the results in this paragraph. The statistical approach for enrichment analysis is described in the methods, "Differential expression analysis."

Line 264-265 and Fig. 2: So the only hypoxia related gene is HIF 1a – so it would be better to state this throughout the manuscript instead of "hypoxia-related genes" (plural). Was collagenase 3 the only collagen-related gene? I think "collagen-degradation and remodeling gene" would be a more accurate moniker.

Corrected as suggested.

Line 301: The sentence beginning with "While" should perhaps start with "Seven".

Corrected as suggested.

Line 305: So if the "collagen-related" genes were collagenases, and they were more expressed in healthy c.f. wasting stars, this would mean that collagen is being broken down, potentially, more in healthy stars than wasting stars. If there are more genes within this ontology – for example, collagen genesis, then I'd recommend splitting these out to reflect constructive and destructive processes.

This is a good suggestion that we explored, however, given that only 4 out of the 51 genes were related to destructive processes, we would not have the power for an enrichment test. Thus, we kept the group as is, but added the details (4/51 destructive) to the methods and added the phrase, "primarily collagen genesis genes" to the results. One of the four is the collagenase gene presented in Fig. 2B, and as predicted it shows higher expression in wasting than visibly healthy animals.

Line 310: "genes related to collagen production" – ah, so this includes both production and destructive processes. I would recommend splitting these. Are these genes all echinoderm transcripts or might these include bacterial transcripts as well? Even with polyA selection, there is always some contamination with prokaryotic transcripts within transcriptomes.

As noted above, they are primarily constructive (47/51), collagen genesis genes, thus we've kept them together. As a manually annotated list, these are all echinoderm transcripts.

Lines 308 – 316: The comparison between healthy and late-stage disease that shows an

upregulation of immune genes is not surprising, since at this stage the specimens are infiltrated by bacteria and other microorganisms through lesions.

What would really help in this list of up- and down-regulated transcripts could be a conceptual figure showing a timeline of disease progression with the various categories.

Please see the new supplemental Table S1 and also Figure 1 of Lloyd and Pespeni 2018 Scientific Reports (also below) which shows the disease progression across the two-week experiment.

Figure 1. Sea Star Wasting Disease progression through the two-week experiment. (A) Photographs taken from one *P. ochraceus* individual as it progressed through SSWD. Numbers in the top left corner of each picture relate to the *P. ochraceus* SSWD symptom guide from seastarwasting.org with the addition of category 5 for dead individuals. (B) Proportion of the 37 individuals of each symptom number at the six sampling time points.

Line 331: While it is certainly interesting to see trends in microbiome composition and correlations to host transcripts, there could well be correlation between un-related processes. Add to this the difficulty in quantitative interpretation based on RNA libraries, and it is this reviewer's opinion that this part of the manuscript does not add much to knowledge of disease process.

Given the revisions in the manuscript, we hope this reviewer finds these results more valuable. As noted above, we would consider removing the 16S data from the study in another revision (can be easily done), but view it as a unique analysis that could spur the field to further and better explore the interactions between hosts and microbes.

Line 358: The last sentence of this paragraph is not complete

Deleted. We performed an additional analysis using Discriminant Analysis of Principal Components (DAPC) that showed the same results as the outlier analyses. We removed the redundancy.

Line 368 – 369: Sea star wasting disease has never been shown to be highly transmissible.

Corrected.

Line 372 – 375: Decreases in relative abundance during disease progression could relate to increases in absolute abundance of other groups – i.e. they may just grow more slowly. It is not appropriate in this reviewer's mind to attribute these as "healthy" or "beneficial" microbiome constituents without rigorous testing to compare health states with and without these organisms. At the very least, quantitative (vs relative i.e. amplicon libraries) approaches are necessary to examine their dynamics during disease progression before these can be identified with any trait.

We agree and have wholly rewritten this paragraph and as noted above removed all mention of "beneficial" or "pathogenic" microbes in the manuscript.

Line 385 – 387: It is not exactly a hypothesis that there are some microbiome constituents correlate with patterns of gene expression in the host.

Corrected.

Lines 400 – 410: Much of this is a restatement of the results. It would be more appropriate to focus on the authors interpretation (i.e. lines 408 – 410), and place these at the start.

This paragraph was cut to half the length.

Lines 434 – 450: Most of this section is speculation. Sure, there are species within the genera identified that can produce e.g. antibiotic and antimicrobial compounds, or may be pathogenic, but extending this to the broad number of taxa within a single genus is highly tenuous.

We agree; we toned down the language but left the ideas as this is the discussion, and added an important caveat that further metagenomic and functional studies are needed. "However, complete metagenomic sequencing and functional studies are needed to identify the specific microbial taxa and the genes in their genomes that may be eliciting a reaction in the host."

Line 454: Unfortunately, the evidence for a healthy microbiome implicated in constantly healthy specimens is not warranted by the data provided (alone).

We agree. We have deleted that statement and revised the title of the manuscript.

Referee: 2

Comments to the Author(s)

Thirty eight asymptomatic *Pisaster ochraceus* seastars were flown from Monterey California to Burlington Vt and held in 1 gallon containers w artificial seawater (3 water changes/week). IN a 15 day lab observation, 16 stars became sick and 8 remained healthy. What happened with the other 14 that were shipped? Were they dead or sick on arrival, suggesting the transit was stressful? IN the Results, line 1 it says: During the two-week experiment, 29 individuals showed signs of SSWD and 8 remained healthy.

All sea stars arrived healthy suggesting that transit was not stressful. We added details to the methods on the transit. The second and third sentences of the results elaborate to show that only 24 of the original 37 sea stars were used for host RNAsequencing (due to the much higher cost of eukaryotic RNAsequencing). We rewrote these sentences to improve clarity. All samples from all days were sequenced for 16S (in the Lloyd and Pespeni 2018 *Scientific Reports* study).

The surface microbiome and immune related genes were monitored in 24 of the stars for 15 days. Why was the experiment terminated at 15 days? Because more and more stars were becoming sick and dying? Inferences were made about causative agent and immune function based on comparing whether the outcome was to become sick or remain healthy. Caution is needed in interpreting microbiome trajectories since this is a lab experiment with long-distance shipped animals that might already be stressed by transport and lab conditions. Further caution is needed in inferring cause and effect between immune condition and bacterial surface changes. While the authors conclude that a healthy immune response drives a beneficial microbiome, its also possible that a healthy microbiome is permissive of a healthy immune response.

We have added new analyses comparing Day 0 microbiome patterns, which were generated from biopsies taken 17 hours after field collection and prior to introduction of the sea stars to artificial sea water, to subsequent days. We show only modest changes in community composition on Days 3 and 6 and a return to the original community composition by Day 9 such that Day 0 and Day 9 are not different. Please see the new Fig. 5 and the detailed responses above.

The study concludes that “Animals that remained healthy had an active immune response and a beneficial microbial community while animals that became sick showed evidence of responding to hypoxia and a proliferation of opportunistic microbes that thrive in carbon-rich, oxygen-poor environments.”

We have revised these conclusions away from ideas of a “beneficial microbial community.”

The immune data are the most useful. The authors found higher expression of immune-related, tissue integrity and collagen genes in healthy relative to sick individuals. I think the result of

lower expression of collagen-related genes in sick relative to healthy is interesting and deserves further investigation. The finding of higher expression of hypoxia-related genes in sick relative to healthy individuals is harder to interpret, since this could be an artifact of lab stress conditions.

We have further elaborated on these methods (definitions of collagen-related and changed hypoxia related to HIF-1a) and results.

I find the microbiome data much less useful because these fairly large animals were likely stressed by transport and being held in extremely small containers (1 gallon), w water change 3 times weekly. Microbiomes are very fast-changing and extremely sensitive to environment. Just the transition from field to lab would be concerning, but to transport across country and held in 1 gallon containers I would expect changed microbiomes.

Please see new details in the methods and new analyses on the effect of lab conditions on microbiome composition, using the 8 healthy animals to test for such effects. As noted in response to Reviewer 1 above, we are willing to remove the microbiome data if reviewers and the editor continue to not see value of these data in the revised manuscript. It would not be difficult to remove and the results would still stand; however, we think it's a novel analysis that will encourage others to test for interactions between hosts and microbes.

A lot of work went into the transcriptomics and microbiome methods and these methods seem adequate. The problem is that the handling of the animals precludes robust interpretation of these results, so this work is not up to the quality I expect for a Proc B paper.

Our new analyses suggest that the handling of the animals little to no effect on the health of the animals or microbiome composition. We believe the important transcriptomic results and unique integration with microbiome data from the only longitudinal study, to our knowledge, of sea star wasting disease, a disease with still unknown etiology, make this study of interest and quality expected for *Proceedings B*.

Based on its limitations, this lab study offers no new insights into the causes of SSWD, although it does try to comment on causes of SSWD. Its main strength is the transcriptional results on immune function. From this lab study, it is precarious and misleading to try and make the following conclusions:

“Taken together, these results suggest that the cause of SSWD may not be a single pathogen as these and other results suggest [17,29], but rather maybe due to a positive feedback loop between abiotic and biotic drivers, where a disturbance of the microbial community, low oxygen conditions, and proliferation of copiotrophic microbes creates an anoxic microenvironment for sea stars leading to hypoxia and loss of collagen control leading to tissue degradation which may further spillover, creating high organic matter, low oxygen conditions for nearby stars [50].

We agree that the concluding statement was not appropriate and it has been replaced with the following: “While the cause(s) of SSW in the wild remain unknown, these results suggest a compromised immune system and the proliferation of specific microbes could contribute to the progression and transmission of sea star wasting disease.” However, as the only time course study of gene expression in sea stars and the study with the most sea stars sampled for RNAsequencing (28), this study offers the most in depth look at sea star gene expression through wasting.

These conclusions would suggest that only stars in hypoxic conditions died from SSWD, and not what occurred: a coast-wide epidemic including extremely pristine, high wave energy environments.

Thank you for this important point. We have removed mention of hypoxia in this sentence.

-----END OF RESPONSE TO REVIEWERS-----

Appendix B

Reviewer(s)' Comments to Author:

Referee: 2

Comments to the Author(s).

The goal of this study was to compare the gene expression patterns of one species of seastar (*Pisaster ochraceus*) affected by the catastrophic sea star wasting disease epidemic. The authors measured gene expression longitudinally of 24 sea stars (*Pisaster ochraceus*), collected from Monterey and shipped to culture in an artificial seawater system in Vermont. 8 individual stars remained asymptomatic and sixteen progressed through stages of SSW over 2 weeks. The rapidity of disease onset in two thirds of the test subjects suggests they were stressed by some combination of shipping and subsequent culture conditions.

Pespeni and Lloyd conclude from longitudinal gene expression patterns that asymptomatic animals remained healthy using an active immune response and that both early- and late-stage wasting animals lost the regulation of systems that maintain tissue integrity.

Their assembled transcriptome, genes of interest, and sequence data files make for useful resources for future work related to Sea Star Wasting. Particularly interesting is the result that previously identified immune, tissue integrity, and pro-collagen genes were more highly expressed in asymptomatic relative to wasting individuals. In wasting stars, hypoxia-inducible factor 1-alpha and RNA processing genes were more highly expressed. It is noteworthy that the expression data indicate compromise of the asteroid catch collagen system, followed by lesions and loss of tissue integrity.

We appreciate the thoughtful, thorough, and favorable review. We provide point-by-point responses below.

The transcriptomic work and analyses are done with an adequate sample size. There is high variability in the data as seen in the large data range and error bars in the longitudinal data plots Fig 4C. I can't tell if this level of variability requires any different statistical handling.

We have removed panels B and C from figure 4 because we realized we did not refer to them in the manuscript and we are near exceeding the length limitations. The variability in those genes and microbes was because they were identified as having correlated abundances with each other (WGCNA module) and the module was identified as being associated with disease phenotype. These genes and microbes were not directly identified as differentially abundant between disease phenotypes (e.g., Figures 1 and 2, DESeq2).

There are some weaknesses in the study:

It is likely that epidemics of aquaculture species, like shrimp viruses (and perhaps salmon viruses) would be considered larger panzootics than SSW, so it does not seem correct to say on

Line 32 this is the largest marine epidemic... It is likely to be correct to follow others in specifying it is largest panzootic of marine wildlife.

Corrected as suggested.

It seems inappropriate and incorrect to refer to non-diseased stars as healthy under these conditions. It is not correct to call the asymptomatic animals healthy, since nothing is actually known about their health state. Asymptomatic as compared to symptomatic would better reflect the differences between stars that developed signs of disease and those that did not.

We agree; corrected as suggested throughout and further defined. Except where we write “apparently healthy” or “visibly healthy” and define those in the methods as asymptomatic

What was the final outcome for these animals after the 2 week study period? Did the 8 “asymptomatic” stars become sick or die? This gets back to concern that its not correct to call these animals “healthy”, all that can be observed is that they are asymptomatic. I regret if I missed seeing this information, but did you report the level of mortality or arm loss in the different treatments?

We ended the experiment at 15 days and froze the eight asymptomatic stars and the remaining wasting individuals to save for potential future research questions.

One gallon containers with no active water flow seem very small for *Pisaster ochraceus* unless these were v small individuals? What were average diameters of the test animals? I am amazed that so many survived in small containers with water changed only every 3 days... The combination of stressful shipping and this kind of lab culture does concern me about the greater relevance of the gene expression results. It seems likely that these kind of stressful conditions explain why two thirds of the test animals developed wasting.

We agree that 1-gallon containers are small for *Pisaster ochraceus*. This comment made me revisit the volume. The containers were actually almost 2-gallon containers and we added 6.5 liters of sea water. The text has been updated accordingly. The stars were average size for *P. ochraceus*; the mean length from tip of ray to the middle of the oral disc was $R = 9.7 \text{ cm} \pm 2.2$ standard deviation, as reported in our previous study as cited (Lloyd & Pespeni 2018). We would have liked to maintain each individual star in a larger aquarium; however, we were limited by space given the large number of animals needed for statistical power and the need to maintain them in individual containers. The results we added to the last version of this manuscript comparing the microbiome on arrival to Day 3 suggest that the overnight journey did not affect the microbiome. Unfortunately, we do not have gene expression data from collection or on arrival to learn more about the physiological state of the sea stars. However, there was no evidence of stress, i.e., no spawning or loss of turgor pressure. We note this an important area of future research.

It's important that for this paper it is made clear that these results are very specific to their

scenario and conditions and these results may not be applicable to what's happening in nature or even in lab-held animals kept under better conditions.

Important point. We have added the following text to the discussion, “Though tracking changes in gene expression and microbiome composition of individual sea stars in the wild is unfeasible, it is important to note that these results may be specific to our laboratory conditions and may not represent the changes occurring during wasting in the wild. However, a study comparing the microbiomes of field-sampled naïve, exposed, and wasting *Pycnopodia helianthoides* sea stars, in lieu of longitudinal sampling of individuals, in Alaska when the panzootic first reached the area reveal a proliferation of anaerobic microbes with exposure and further with wasting [57], further supporting the idea that low oxygen may play an important role in SSW [19].”

Please specify what species of sea star Gudenkauf and Hewson used in their transcriptomic study.

Pycnopodia helianthoides is noted in the text as “In the same species, ...” Give length restrictions, we didn’t state the species again.

a few typos/extra words:

Thank you for catching these.

o Line 266: period after "healthy than." is not needed

Corrected.

o Lines 277 and 292: "collengenase" should be "collagenase"?

Corrected.

o Line 333: don't need the word "in" before the word "(orange)"

Corrected.

Referee: 3

Comments to the Author(s).

This paper provides critical information to help us understand a dramatic and barely understood marine epidemic, the largest marine wildlife epidemic ever documented. The organisms involved, sea stars, have historically received remarkably little research attention, despite their critical roles in ecosystems, their phylogenetic affinity as a deuterostomes, and their enduring popular appeal. Experiments with sea stars and wasting disease are exceedingly few and the microbiome is an excellent if challenging place to look for disease relationships. The paper is greatly improved with the critical eyes of reviewers' and editor's many suggestions that have clarified and highlighted what is known and not known. It is of further great interest how the authors carried out the lab microbiome experiments without seeing substantive control treatment changes as well...that is also well worth reporting as they have done in the revision. A little further info here would be helpful. Specifically, where in particular were the

stars collected in Monterey? This would be helpful background info for future comparisons and for knowledge about how that particular population has been faring with SSWD, in general. Also, I assume the stars were not fed during the experiment, but perhaps that should be stated.

Thank you for your supportive review. In addition to the GPS, we have added the location details “off the Monterey wharf” to the present manuscript. Correct, the stars were not fed to avoid introducing microbes or differences among the stars/aquaria. This has been added.

-----END RESPONSE TO REVIEWERS -----

Appendix 7

Reviewer(s)' Comments to Author:

Referee: 4

Comments to the Author(s).

This manuscript explores the gene expression of *Pisaster ochraceus* in the lab, comparing individuals that spontaneously wasted following collection and transportation to lab conditions to those that remained asymptomatic across the same conditions. Within the spontaneously wasting group, gene expression was compared between different stages of wasting. In addition, the manuscript compares gene expression and microbiome abundance correlations within the same dataset. The authors found that there was differential gene expression throughout the compared stages of wasting, with asymptomatic individuals showing increased expression of genes related to tissue integrity, collagen and immune responses, and symptomatic individuals showing increased gene expression of RNA processing genes and hypoxia related genes. They found that microbiomes associated to asymptomatic individuals remained relatively stable (with some changes over time), and that there were some microbial species that were associated with symptomatic individuals. The authors additionally found no genotype variants that were associated with symptomatic status in the experiment.

I think this was a clearly written manuscript, with elegant figures that provides important information on a relevant and interesting question. There were a few minor points that I think could be addressed, including a few typos, which are outlined below.

We sincerely appreciate the thoughtful, thorough, and favorable review.

Likely while this manuscript has been in review, and additional manuscript was published that looks at wild *P. ochraceus* microbiomes and it would be worth incorporating some short points from Loudon et al, 2023 into the discussion.

Loudon, A. H., J. Park, and L. W. Parfrey. 2023. Identifying the core microbiome of the sea star *Pisaster ochraceus* in the context of sea star wasting disease. *FEMS Microbiology Ecology*.

We have added this important new work to the discussion.

Line edits:

Line 98: This is a minor point, but stating that Monterey 'in the center' of the species range seems like a pretty broad statement. Since the range is Baja to AK, it would depend on how wide a range you consider 'the center' whether that statement is true. Can you either soften the language or cite the sources for Monterey being considered the center?

Language softened to “middle section.”

Line 188: I think there is a word missing here

Corrected missing word – added “between.”

Figure 1D: why did you choose to combine all samples here, and not split between asymptomatic and wasting, as is done in the other panels? Especially since the statistic is about the comparison between wasting and asymptomatic, not between collagen and non-collagen expression.

This is a good question/observation. I believe the reason why is because when we initially ran the analyses, we did so by classifying as collagen vs. non-collagen based on the identified gene list and not by the additional factor of health status.

Line 311-312: I am not sure why ‘in the common environmental conditions of individual lab aquaria’ is included in this statement of the result. It seems odd to point that out here, when it is likely true of all other results as well.

Deleted this clause.

Line 464-465: ‘cell adhesion’ is currently referenced twice.

Thank you – deleted.

Line 484: Missing a word

We believe we added the missing word identified.

Line 517-518: How sure can you be that all stars in your experiments were exposed to wasting? Were they ever co-housed while some stars showed signs of wasting? Are you assuming they were all exposed because they were collected from the same location? This is a different framing than has been used through the manuscript and makes a claim that is a bit beyond what you currently have support for.

Thank you for making that observation of the error of our language. We have revised to read “while other collected from the same environment waste.”

Referee: 3

Comments to the Author(s).
Changes have improved the ms.

A few further small notes for clarification in the methods:

Pg 3, line 40: not reduction in habitat, but rather, alteration in habitat

Good catch – corrected as suggested.

Pg 5, line 89: “the wharf at Monterey Harbor”. Please be more specific, by telling us which wharf, the depth range, and something about the habitat

We have added these details about the collection location.

Pg 6-7, what criteria were used to select the particular 24 out of 38 stars?
What was the size range of the stars?

We have added the size range to the paper (was noted in the original study). We added “all those that remained asymptomatic and a random selection of those that wasted” to report the selection criteria.

How deep is the biopsy punch? Does it go all the way through the body wall and into the coelomic cavity? Punch into organs?

We added the following sentence, “Biopsy punches were about the size of half of a grain of rice and only included epidermal tissue.”